# Distinct shared and compartment-enriched oncogenic networks drive primary versus metastatic breast cancer

Zhe Jiang[1], YoungJun Ju[1,14], Amjad Ali[1,14], Philip E. D. Chung [1,2,14], Patryk Skowron[2,3,4,14], Dong-Yu Wang [1,14], Mariusz Shrestha [1,2], Huiqin Li[1], Jeff C. Liu[5], Ioulia Vorobieva[1,2], Ronak Ghanbari-Azarnier[1,2], Ethel Mwewa [1], Marianne Koritzinsky[6], Yaacov Ben-David [7,8], James R. Woodgett [9], Charles M. Perou [10], Adam Dupuy [11], Gary D. Bader [5,12], Sean E. Egan [3,12], Michael D. Taylor [2,3,4] & Eldad Zacksenhaus [1,2,13] ✉

Metastatic breast-cancer is a major cause of death in women worldwide, yet the relationship between oncogenic drivers that promote metastatic versus primary cancer is still contentious. To elucidate this relationship in treatment-naive animals, we hereby describe mammary-specific transposon-mutagenesis screens in female mice together with loss-of-function *Rb*, which is frequently inactivated in breast-cancer. We report gene-centric common insertion-sites (gCIS) that are enriched in primary-tumors, in metastases or shared by both compartments. Shared-gCIS comprise a major MET-RAS network, whereas metastasis-gCIS form three additional hubs: Rho-signaling, Ubiquitination and RNA-processing. Pathway analysis of four clinical cohorts with paired primary-tumors and metastases reveals similar organization in human breast-cancer with subtype-specific shared-drivers (e.g. RB1-loss, TP53-loss, high MET, RAS, ER), primary-enriched (EGFR, TGFβ and STAT3) and metastasis-enriched (RHO, PI3K) oncogenic signaling. Inhibitors of RB1-deficiency or MET plus RHO-signaling cooperate to block cell migration and drive tumor cell-death. Thus, targeting shared- and metastasis- but not primary-enriched derivers offers a rational avenue to prevent metastatic breast-cancer.

Metastatic dissemination is the major cause of death from diverse types of cancers including those of the breast and is the target of emerging therapeutic strategies[1–4]. In the United States for example, 6% of women exhibit metastatic breast cancer at initial diagnosis, 12.7% develop invasive breast cancer each year, and 15% succumb to the disease (https://www.cancer.net/cancer-types/breast-cancer-metastatic/statistics). In contrast, primary tumors, the focus of many therapies, are commonly removed and discarded. The oncogenic relationship between primary and metastatic lesions is therefore critical for judicial therapy, yet it is highly contentious. One model posits that the oncogenic alterations that drive primary tumor growth also propel metastatic dissemination[5]. Indeed, multiple studies show that oncogenes and tumor suppressors that induce primary tumor growth also promote metastasis [reviewed in ref. 6], and this is further supported by the high concordance between the oncogenic landscape of metastases and primary lesions from the same patients [ref. 7. and references therein]. Other reports highlight new alterations in metastatic versus primary lesions[8–15]. Such metastatic-specific alterations may contribute to the cellular plasticity required for the metastatic cascade[6], but may also reflect selective pressure and emergence of rare clones that survive therapy[16,17]. Thus, the exact correlation between oncogenic drivers

that promote clonal evolution at the primary site and those that facilitate metastasis, in the absence of drug treatment, is not fully understood. Elucidating this relationship in drug-untreated subjects is important for understanding the metastatic process and for developing effective therapies.

Sleeping Beauty (SB) transposon-mediated mutagenesis enables the identification of oncogenic drivers that promote primary and metastatic cancer in therapy-naive mice[18–24]. Gene-centric common insertion sites (gCIS) can be detected by ligation-mediated PCR and deep sequencing from minute or even microscopic amounts of metastases. SB screens offer several additional advantages: tumor formation in immune-competent mice; acceleration of metastasis in cooperation with oncogenic drivers that usually do not suffice to promote visible metastases; and simplification of detection of all genetic and epigenetic alterations (e.g. mutations, deletions, insertions, amplifications, DNA methylation, chromatin modifications) through a single platform, namely, insertional transposition. As noted, while many oncogenic alterations found in metastatic cancer in humans reflect drug-resistance, SB mutagenesis in mice can identify actual metastatic drivers in the absence of drug selection. SB mutagenesis screens have been carried out in multiple tissues including the mammary gland and brain on the background of different oncogenic drivers including p53 mutation, Pten-loss, Brca1-loss, or PI3K and Notch gain-of-function[25–28]. Only a subset of screens was extended to the metastatic niche[12,29,30], revealing minimal overlap between the two compartments, and only one report, on hepatocellular carcinoma, demonstrated clonal relationships between primary and metastatic insertion sites[23].

Deregulated cell proliferation, a major hallmark of cancer, often involves the disruption of the retinoblastoma (RB) tumor-suppressor gene[31,32]. In breast cancer, this is often achieved by genetic/epigenetic ablation (mutations/deletions/promoter-silencing) of RB, or by hyperphosphorylation of the protein, pRB, through activation of cyclin dependent kinases CDK4/6 and CDK2[33]. Indeed, it is estimated that RB is disrupted in 20–40% cases of TNBC[34–37]. RB pathway loss, alongside TP53-pathway disruption and induction of PI3K signaling, is also a major driver of metastatic breast cancer and other malignancies[9]. We previously demonstrated that conditional deletion of the murine Rb gene in the mammary epithelium, using a floxed Rb allele and MMTV-CRE[NLS] transgenic mice, induces diverse mammary tumors, including luminalB-like and basal-like breast cancer as well as HER2-like lesions[36,38,39]. To identify oncogenic networks that cooperate with loss of Rb/cell cycle control to promote primary and metastatic breast cancer, and determine the relationship between the two compartments, we herein performed a mammary-specific SB mutagenesis screens on an Rb-deficient background.

In this work, we identify primary (P-) and metastasis (M-) drivers as well as drivers that promote both primary and metastasis, which we termed shared (S) oncogenic drivers. Our results suggest that S-drivers cooperate with distinct local networks to promote primary versus metastatic growth, and reveal similar organization in human breast cancer with shared oncogenic drivers, primary-enriched and metastasis-enriched pathways, with direct implications for the prevention of metastatic disease.

## Results

### Sleeping beauty mutagenesis screens identify shared, primary- and metastasis-specific drivers

To identify genes that cooperate with dysregulation of the cell cycle to promote tumorigenesis, we performed SB mutagenesis screens in combination with Rb deletion, using a floxed Rb exon-19 allele and MMTV-Cre[NLS] transgenic deleter mice (Fig. 1a). We previously showed that multiparous MMTV-Cre:Rb[f/f] mice develop diverse mammary tumors as well as microscopic lung lesions presumed to represent metastatic descendants from primary mammary tumors[38]. The MMTV-Cre:Rb[f/f] mice were crossed with an activatable SB11 transposase, knocked into the ROSA26 locus (R26[lsl_SB11]), and two different transgenic transposon concatemers, T2/Onc3a and T2/Onc3b, containing 11 and 28 copies on chromosomes 9 and 12, respectively[19,40,41]. As local hoping within the same chromosome occurs at higher frequency than across chromosomes, usage of both transposon donor lines enables coverage of the entire genome by omitting local transpositions in the T2/Onc3a and T2/Onc3b screens. The T2/Onc3a and b transposons contain a strong synthetic promoter, CAG, designed for efficient epithelial cell-based screens.

Both MMTV-Cre:Rb[f/f]:T2/Onc3a:R26[lsl_SB11] and MMTV-Cre:Rb[f/f]:T2/Onc3b:R26[lsl_SB11] female mice developed mammary tumors as well as large, macroscopic pulmonary lesions that could readily be dissected under a stereomicroscope (Fig. 1c). DNA extracted from primary and lung tumors were subjected to PCR to test for deletion of the Rb floxed allele following CRE-mediated recombination. Not all lesions showed Rb deletion (Fig. 1d), indicating Rb-dependent and -independent induction of mammary tumors via SB-transpositions. Thus, we used PCR to test for Rb deletion in all primary and lung lesions, and used samples with a confirmed Rb null genotype in subsequent experiments. Primary tumors exhibited diverse histology including pleomorphic/squamous cell carcinoma (P/SCC) and adeno-squamous carcinoma (ASC), which were most prominent, as well as papillary/micropapillary (P/MP), poorly differentiation adenocarcinoma (PDA) and cribriform (CF) (Fig. 1d, e). Lung lesions also showed highly heterogeneous histology as seen in primary tumors, with most frequent subtypes being PDA and P/MP. Notably, the macroscopic lung lesions were used for DNA extraction and therefore the histology of metastases represents lesions from the remaining lung samples.

To identify genes affected by SB transposon insertions, we subjected 116 mammary tumors and 79 lung metastases, all with confirmed Rb deletion, to ligation-mediated PCR with 79 bar-coded primers, followed by next-generation DNA sequencing and bioinformatic analysis. Specifically, we prepared SB insertion libraries using an unbiased Shear-SPLINK method, which uses sonication rather than restriction enzymes to fragment the DNA[42]. We subsequently sequenced and filtered the reads to include only common insertions starting with a TA dinucleotide characteristic of a canonical SB insertion site. Using gene-centric common insertion site (gCIS) analysis[43], we were able to robustly call genes driving growth of mammary and lung tumors. We tabulated gCIS at both the highly stringent filtered_clonal (Supplementary Fig. S1a, Supplementary Data S1 and 2 with legends in Additional Supplementary File) as well as the filtered_subclonal levels (Supplementary Data S3 and 4). The top 15 and 23 primary and lung filtered_clonal gCISs, respectively, are shown in Fig. 1f. Altogether, 80 and 85 statistically significant gCISs were observed in primary sites and lungs, respectively, with an overlap of 7 shared genes (Met, Prlr, Nf1, Jup, Map3k3, Stat5b and Notch1), henceforth referred to as S-drivers (Fig. 1g). SB transposon insertions in the 7 S-drivers predict inactivation for NF1 and activation for the rest. As examples, for NF1 there were multiple transposon insertions in the same or opposite orientation relative to the gene in both primary tumors and metastases, suggesting loss-of-function through truncation/early termination (Fig. 1h), whereas for cMET, most insertions were in the same orientation as gene transcription, and mapped within the promoter region or intron 1, consistent with transcriptional activation (Fig. 2a).

Of the 79 metastatic biopsies subjected to gCIS analysis, we identified 56 filtered_clonal gCISs while 23 samples had no significant gCIS (Supplementary Data S1 and 2). The 7-drivers were found in 40 of the 56 samples with gCIS (71.4%). The remaining 16 samples (28.6%) are driven by different or functionally similar oncogenic networks.

The primary- and metastatic-specific gCIS partially overlapped with known oncogenic landscapes of primary and metastatic cancer

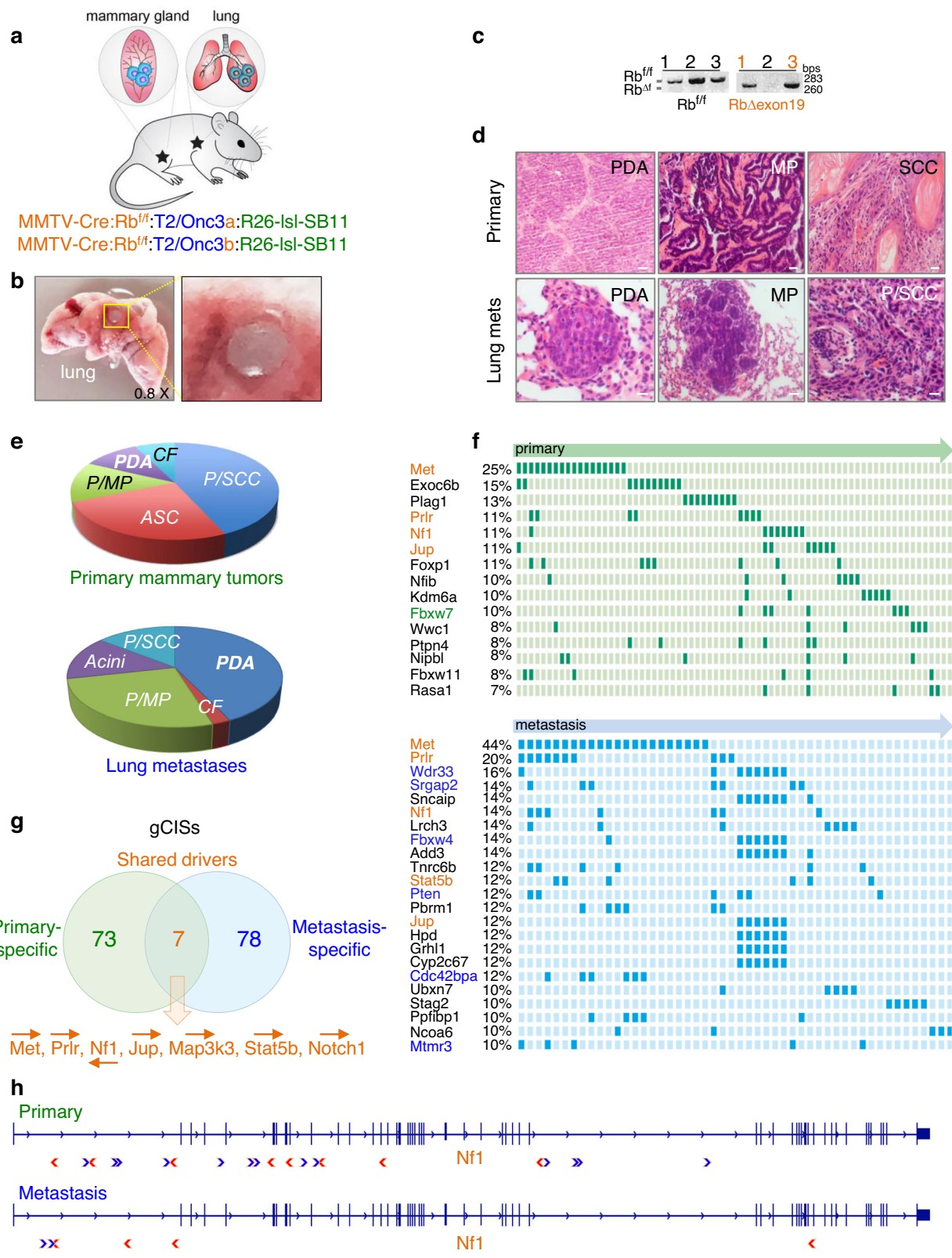

(Supplementary Fig. S1b–e). Indeed, some of the primary (P-) gCIS we identified, including *Nf1, FoxP1, Kdm6a, Fbxw7, Smad4, Notch1, Met, Chd1, Fat1* and *Erbb2* as well as *Rb* and its upstream regulators (*CCND1, CCNE1, CDKN2A, CDKN2B, CDK6, CDKN1B*), are altered in primary and metastatic human breast cancer[35,44], whereas S- and metastatic (M-) gCIS: *Met, Nf1, Map3k3, Pten, Asxl2, Pten, Pbrm1, Notch1* and *Rb*, are altered in metastatic breast cancer[9]. In all, six of

the seven S-drivers (*Jup* excluded) have known alterations in primary and metastatic breast cancer, highlighting the importance of the oncogenic pathways they induce; JUP/Plakoglobin/gamma-catenin plays a critical role in collective migration of disseminating tumor cells and in breast cancer metastasis, and its expression correlates with decreased distant metastasis-free[45] and overall survival (OS; *P* = 0.0007; Supplementary Fig. S2a).

**Fig. 1 | Sleeping Beauty (SB) mutagenesis screens on Rb-deficient background identify overlapping oncogenic alterations in mammary and lung tumors.**
**a** Setup of SB screens using MMTV-Cre:Rb^f/f^:T2/Onc3a:R26-lsl-SB11 and MMTV-Cre:Rb^f/f^:T2/Onc3b:R26-lsl-SB11 mice to identify primary mammary tumors and lung metastases driven by SB transposons on Rb-deficient background.
**b** Macroscopic lung tumors detected by stereomicroscope were dissected and subjected to ligation-mediated PCR and deep sequencing. **c** PCR-based analysis of primary tumors to identify the 283 bp Rb-floxed allele (lanes 1–3) and 260 Rb deletion product following Cre-mediated recombination (lanes 1 and 3, but not 2). Source data are provided as a Source data file. Representative histology (**d**) and pie distribution (**e**) of primary and lung lesions. Pleomorphic/Squamous cell carcinoma (P/SCC), Adenosquamous carcinoma (ASC), Papillary/ Micropapillary (P/MP),

poorly differentiation adenocarcinoma (PDA), Cribriform (CF). Scale bar, 100 µm. **f** Top gene-centered Common Integration Sites (gCIS) identified in primary (top) and lung (bottom) tumors following SB screens on Rb-deficient background. All gCISs are listed in Supplementary Fig. 1A and shown in Supplementary Data 1 and 2. S-specific gCIS are highlighted in red, and percentage in primary and lung lesions is tabulated. Representative gCIS functionally analyzed herein are marked in green (top – Fbxw7) and blue (bottom – Wdr33, Srgap2, Fbxw4, Pten, Cdc42bpa, and Mtmr3). **g** Venn diagram for significant gCISs identified in primary and lung lesions and the 7 S-drivers. Arrows point to direction of transposons. **h** Schematic structure of the NF1 gene locus and relative location of CIS in primary tumors and lung metastasis. > denotes SB integration in the 5′ to 3′ direction of the gene; <denotes reverse direction.

## Clonal relationship between mammary tumors and lung metastases

Before any further analysis of the gCISs identified above, we sought to address the clonal relationship between mammary and lung tumors identified with the same animal. This is a critical issue as lung lesions may not represent bona fide mammary tumors that have metastasized to the lung, but independent primary lesions induced through cryptic induction of CRE-like activity in the germline or lung epithelium. Indeed, ubiquitous immobilization of the transposase has been shown to induce different tumor types including pulmonary adenoma and carcinoma[41]. Germline inactivation of *Rb* is improbable because homozygous loss of this tumor suppressor is embryonic lethal[46,47], but specific immobilization of the transposase and loss of Rb in the lung, leading to primary pulmonary tumors could not be excluded a priori.

To test for clonal relationship between mammary and lung gCIS, we took advantage of the exact sequencing data of transposon-integration sites to compare primary and metastatic lesions from the same animals, using filtered_subclonal data. Examples of such analysis are shown in Fig. 2b and with more detail in Supplementary Data S5. In the case of cMET, there were 5 mice with primary and lung tumors, both containing insertions in the gene on chromosome 6. For mouse 31 (pair #1), 4 different SB transposons were identified, one of which was integrated in the first intron of the *cMET* locus between nucleotides 17436202 and 17436203. This same integration site from the primary lesion was found in 9 different lung tumor samples from the same mouse, demonstrating clonal relationship (Fig. 2b–d; Supplementary Data S5). The three other clones in this mouse with SB transposons in *cMET* either did not give rise to macro-metastasis or such micro-lesions were not detected by deep DNA sequencing analysis (Fig. 2d). A second mouse (#2) contained a single transposon in its primary tumor (17408640–17408641 at the 5′) that was also found in a single metastasis from the same mouse; mouse #3 had two different transposons, one of which (17413148–17413149 at the 5′) gave rise to a lung metastasis with identical integration; and mouse #4 had three different transposons, one of which (17439278–17439279 in intron 2) gave rise to a metastasis with identical integration. This mouse also contained another metastasis with transposon integration in intron 1 (17414760–17414761), which likely arose from a different area of the primary tumor that was not subjected to gCIS analysis. Finally, a fifth mouse (#5) had a lung metastasis that was not related to the primary tumor (Supplementary Data S5), and was again, likely derived from another area in the primary tumor that was not subjected to deep sequencing (Fig. 2d).

Similar clonal relationships were found for *Jup, Map3k3, Fbxw4* and *Plag1* in one mouse each, and for *Stat5b* in two mice (Fig. 2c; Supplementary Data S5). Thus, lung lesions are clonally related to the primary mammary tumors in all mice in which such analysis could be performed and therefore, by extension, in the entire cohort.

As noted, this analysis was done by examining gCIS at the filtered_subclonal level. As such, insertions in *Plag1*, found at the filtered_clonal level to be primary-only (Fig. 1f; Supplementary Data S1),

were detected at the filtered_subclonal level in lung metastases, whereas insertions in *Fbxw4*, found only in lung metastases at the clonal level, were also detected in primary lesions at the subclonal level (Supplementary Data S3 and 4). Therefore, while our designation of P- and M-specific gCIS is based on the highly stringent filtered_clonal analysis, at the subclonal level, they can be denoted P- and M-enriched gCIS.

## A major network of shared drivers, and metastatic-specific hubs
String analysis for protein-protein interaction among gCISs in primary tumors revealed a single major network centered around cMet that included Nf1, Erbb2 and Notch1 as well as downstream proteins (Fig. 2e, left). String analysis of gCISs in the metastatic compartment revealed a more interconnected network around cMet, comprising Nf1, HRas, Stat5a-Stat5b, Notch1, Pten, and Grb2, each associated with multiple other proteins (Fig. 2e, right). In addition, the metastatic compartment contained three smaller hubs (demarcated in colors), components of which have been implicated in cancer progression and invasion in other malignancies: (i) Rho signaling/cell migration genes such as SRGAP2 (SLIT-ROBO Rho GTPase Activating Protein 2)[48], CDC42BPA (CDC42 Binding Protein Kinase Alpha; also known as Myotonic Dystrophy Protein Kinase-Like Alpha – MRCKA)[49] and WASF2 (WAVE2)[50]; **(ii)** ubiquitination pathway enzymes such as FBXW4[51], UBXN7, SPOP[52], HECTD1[53] and UBE2D3[54]; and **(iii)** pre-mRNA processing factors including WDR33, involved in cleavage and polyadenylation of pre-mRNA 3′ ends[55], CWC22, CDC5L[56], PRPF6[57], and PSPC1[58].

To identify signaling pathways enriched by gCIS in each compartment, we performed G:Profiler analysis[59]. Gene ontology (GO) for Biological Processes revealed that primary tumors were uniquely enriched in genes involved in 'metabolism', 'signal suppression', 'biosynthetic inhibition', and 'programmed cell death' as well as in 'cell migration', 'epithelium differentiation', 'developmental growth', 'transcription', and 'tube development' (Fig. 3a). The latter five pathways were also the major pathways enriched in the metastatic niche. The specific genes in the 'cell migration' pathway in each compartment are listed in Fig. 3b. All 7 S-drivers were components of the shared 'cell migration' genes in the primary (green) and metastasis (blue) compartments. In addition, metastasis-specific 'cell migration' genes included ATF2, ETS1, HECTD1, PBRM1, PRKACB, PTEN, SRGAP2, STAT5A and WASF2, whereas primary specific 'cell migration' genes comprised FBXW7, ITGB3, KDM6A, PLAG1, RASA1, SMAD4, SOCS5 and TGFBR1. Kaplan–Meier relapse-specific survival (RFS) analysis of genes from the metastasis-specific hubs (Fig. 2e, right) and migration pathway (Fig. 3a) revealed significant impact on clinical outcome (Fig. 3c; Supplementary Fig. S2b, c).

The S- and some of the M-specific gCIS identified in our screens are known to induce cell proliferation, survival and migration (Fig. 3d). The major pathway involves the cMET receptor tyrosine kinase and downstream factors including components of the RAS pathway (NF1, HRAS, GRB2, MAP3K3, ETS1 and ATF2). The Prolactin receptor (Prlr) pathway overlaps with the cMET pathway, with both inducing STAT5, which is targeted in both the primary

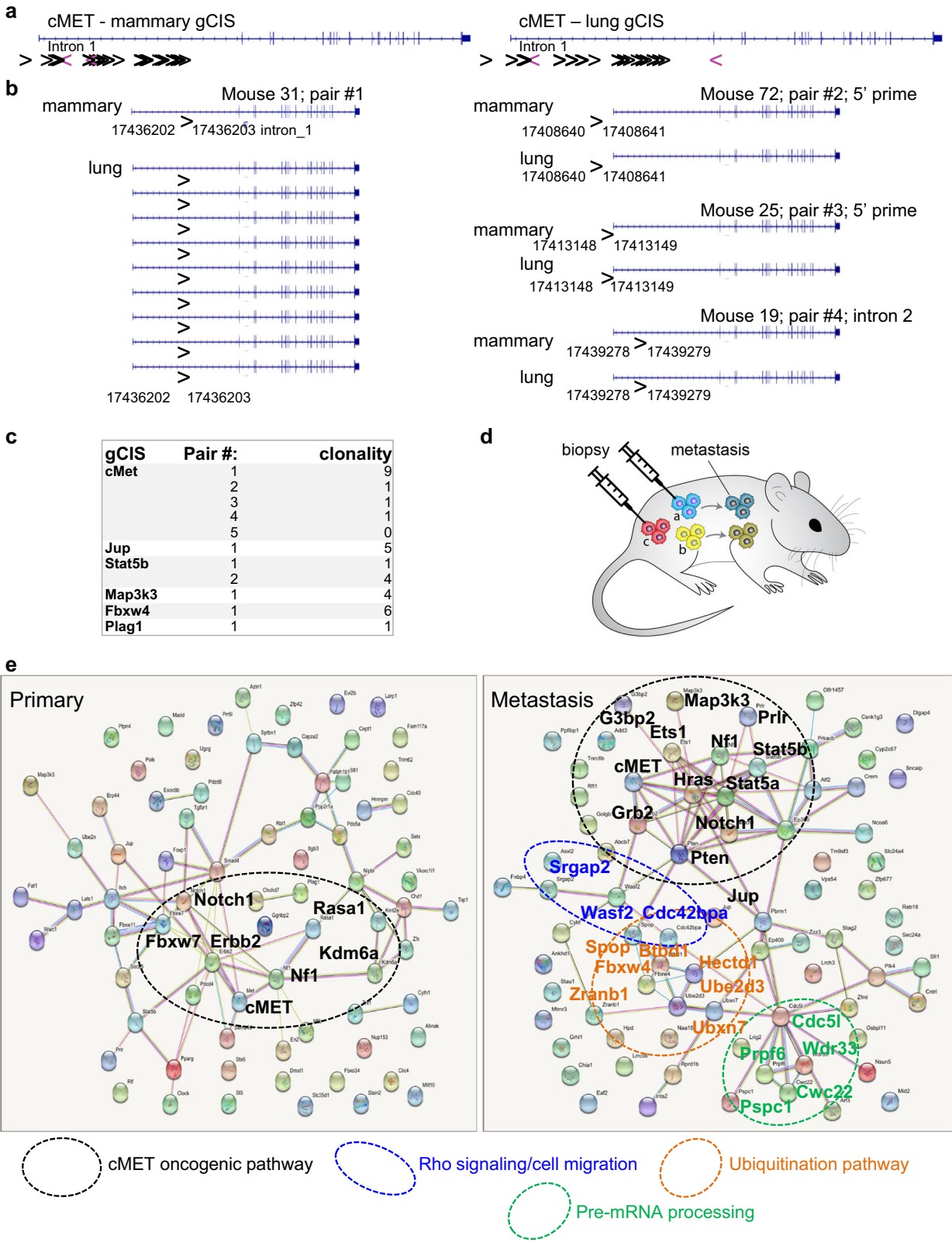

and metastatic gCISs. Another S-driver, Notch1, is implicated in cell proliferation, survival and migration through AKT, PTEN and Rho signaling[60–62].

**MET pathway-high plus RB-loss in patients with poor prognosis**

The MET and RAS pathways are key drivers of primary tumor growth and metastasis, and major targets for therapeutic intervention[63–66].

However, the combined effect cMET or RAS pathway activation plus RB-loss on clinical outcome of breast cancer patients, and the major breast cancer subtypes in which they are altered are not well defined. The cMET proto-oncogene, which is rarely amplified in breast cancer, is over-expressed to a different degree [ref. 67 and references therein], and is further activated by phosphorylation via various paracrine/autocrine signaling[63]. Thus, we employed a cMET signature, developed

**Fig. 2 | Clonal relationship between gCISs in primary and lung lesions and their interactions in each compartment. a** Schematic structure of the cMET gene and relative location of gene-centered Common Integration Sites (gCIS) in primary tumors and lung metastasis. > denotes SB integration in the 5′ to 3′ direction of the gene and <denotes reverse direction. **b** Nucleotide-resolution integration site analysis of SB transposons in primary and lung lesions from the same mice in 4 different animals in the cMET gene. For example, mouse 31 (pair #1) has an identical integration site in a primary tumor and in 9 different lung metastases. Pairs #2–4 have a single lung metastasis each with identical integration site as in the primary tumors. **c** Summary of clonal relationship observed between primary lesions and metastases in 6 different gCISs (details in Supplementary Data 5). **d** A schematic representation of clonal relationship between primary and lung lesions. For gCIS

analysis, tumor biopsies and whole macro-metastasis were subject to ligation-mediated PCR and next-generation DNA sequencing. If a tumor biopsy (**a**) contains a disseminating clone that gives rise to a large metastasis – and deep sequencing detects the same integration sites in both compartments, clonal relationship can be established. On the other hand, if a tumor biopsy (**b**) with disseminating metastatic clone is not analyzed for gCIS, or (**c**) does not spawn a disseminating clone, clonality cannot be demonstrated. **e** String analysis for interaction among gCISs in primary- and metastasis-specific gCIS. Demarcated are cMET hubs found in both compartments (black circles), as well as Rho signaling/cell migration (blue), protein ubiquitination (orange) and pre-mRNA processing (green) hubs identified in metastasis-only gCISs.

for hepatocellular carcinomas (HCC), as a surrogate for its induction[68]. The 24 gene signature (MET24) comprises genes involved in the regulation of oxidative stress response, cell motility, cytoskeletal organization, and angiogenesis, and is highly elevated in metastatic HCC and in primary HCC with poor clinical outcome. When applied to the METABRIC breast cancer cohort, the MET24 signature was most elevated in basal-like breast cancer, and lowest in luminal A and normal-like breast cancer (Fig. 4a). MET24 was also elevated, albeit to a lesser extent, in claudin-low, HER2-enriched and luminal B breast cancer samples. When analyzing all breast cancer subtypes, high MET24 activity significantly but moderately correlated with poor disease-free (DFS) and overall (OS) survival with hazard ratios (HR) of 1.65 and 1.24, respectively (Fig. 4b).

The observed increase in cMET signature activity in basal-like breast cancer (Fig. 4A) prompted us to investigate its impact on specific subgroups of triple-negative breast cancer (TNBC). TNBC comprises at least 6 different subtypes: basal-like 1 (BL1), basal-like 2 (BL2), mesenchymal (M), mesenchymal stem–like (MSL), immunomodulatory (IM), luminal androgen receptor (LAR), and unspecified (UNS)[69]. We previously demonstrated that the largest subgroup, BL1, includes a subset of highly lethal tumors defined by loss of the tumor-suppressor PTEN (Phosphatase And Tensin Homolog; Phosphatidylinositol-3,4,5-Trisphosphate 3-Phosphatase and dual-specificity protein phosphatase) plus five specific microRNAs (labeled red in the PTEN^low/miR^low row)(Fig. 4c)[70,71]. This lethal group also exhibits *TP53* mutation, *RB* loss and high *MYC*, *β-catenin/WNT*, *PI3K* and *RhoA* signaling. The MET24 signature was significantly enriched in BL-1 compared with all other subtypes (Fig. 4d, top), and completely overlapped with the most lethal PTEN^low/miR^low tumors (Fig. 4c, demarcated by red box).

While the MET24 signature was most elevated in BL1, the RAS pathway was induced in BL-1 as well as in BL2 and MSL (Fig. 4d, bottom). In TNBC, a high MET24-pathway signature identified patients with poor OS with hazard ratio of 2.24 ($P = 0.007$; Fig. 4e). Within 72 BL1 TNBC patients, the MET24 signature predicted OS with HR of 2.29 ($P = 0.025$; Fig. 4f). In an independent cohort of 383 TNBC patients[72], high MET24 expression coincided with unfavorable, albeit not statistically significant, outcome; yet, within 100 BL1 patients from this cohort, high MET24 signature defined patients with significantly reduced OS, with HR of 1.79 ($P = 0.041$) relative to MET24-low BL1 patients (Fig. 4g). Thus, MET24 consistently identifies BL1 patients with exceptionally poor prognosis. Contrary to MET24, RAS pathway activation did not correlate with TNBC patient prognosis (Fig. 2e, center), and MET24 high/RAS-high patients exhibited insignificant increase in HR (1.87; $P = 0.1$; Fig. 2e, right) compared with high MET24-only patients (Fig. 2e, left; HR = 2.24, $P = 0.007$). In contrast, MET24 high/RB-loss TNBC patients exhibited worse clinical outcome in three independent cohorts (Fig. 4h; Supplementary Fig. S3a–c). Together, these results reveal that MET pathway activation, which is frequently targeted by gCISs in our SB screens, is predictive of poor clinical outcome of TNBC patients, and that MET but not RAS pathway activation marks highly aggressive BL1 TNBC lesions that are characterized by RB-loss, TP53 mutation and high MYC, β-catenin/WNT, PI3K and RhoA signaling.

As a prelude to our cell culture analysis of gCISs, we assessed MET and RAS pathway activation as well as RhoA signaling in 59 breast cancer cell lines using the Cancer Cell Line Encyclopedia database (CCLE)[73], classified by PAM50 into BasalA, BasalB/Claudin-low, HER2-enriched and luminal-like, as described[74] (Fig. 4i). In many BasalA and BasalB/Claudin-low-like cell lines, either MET or RAS, or both pathways were elevated. Three TNBC cell lines used in our subsequent analysis of gCIS, MDA-MB-231, MDA-MB-436 and MDA-MB-468, as well as the luminal line MCF7, are highlighted in red, whereas those used to determine the effect of RB depletion on migration are marked by asterisks.

## Genetic analysis of combined *Rb-Fbxw7* and *Rb-Pten* deletions in mammary epithelium

Since components of the major cMet hub and the network of downstream S-drivers are well established to promote growth, motility, and dissemination, we next sought to determine whether selected M-specific gCIS from the three other hubs in Fig. 2e also promote cell motility and dissemination. We selected for further analysis CDC42BPA-SRGAP2, FBXW4 and WDR33, which are the most frequently hit gCISs from each of the three hubs (Supplemental Fig. S1a), as well as three additional genes: protein phosphatase PTEN and MTMR3 (Myotubularin Related Protein 3; Phosphatidylinositol-3-Phosphate Phosphatase) found only in the metastatic screen, and the E3 ubiquitin-protein ligase FBXW7 (F-Box and WD Repeat Domain Containing 7), found only in the primary screen. gCIS analysis predicts that all these genes, excluding WDR33, which is induced and CDC42BPA which contains transposons in both orientations, are inactivated by SB insertional mutagenesis. To validate these gCISs, we employed two approaches: genetic analysis in mice using conditional alleles, and gene depletion via small hairpin RNA (shRNA) in cultured breast cancer cell lines followed by analysis in vitro or after orthotopic transplantation into recipient mice.

We first assessed the effect of combined loss of *Rb* and *Fbxw7* in the mammary gland as these two genes are frequently lost in ER-negative breast cancer[34,35,75–79]. For this, we crossed MMTV-Cre:Rb^f/f mice with Fbxw7^f/f mice[80] (JAX Stock No: 017563). Nulliparous MMTV-Cre:Rb^f/f:Fbxw7^f/f and control female mice were harvested at 12–16 months of age, subjected to whole mount staining and large macro-lesions were scored under a dissecting microscope (Fig. 5a). We observed a significant increase in tumor formation in MMTV-Cre:Rb^f/f:Fbxw7^f/f versus non-transgenic control or MMTV-Cre:Fbxw7^f/f ($P < 0.0001$) but only a slight and insignificant increase in comparison with MMTV-Cre:Rb^f/f mice ($P = 0.217$, one tailed *t*-test).

Next, we analyzed the effect of combined mutations in *Rb* and *Pten*, the latter of which was identified in the metastasis-only screen (Supplementary Data S2). Combined deletion of *Rb* plus *Pten* (MMTV-Cre:Rb^f/f:Pten^f/f mice) did not accelerate tumor formation relative to *Rb* or *Pten* loss alone (Supplementary Fig. S4a). Furthermore, analysis of lungs from these mice revealed statistically significant increase in the number and area of lung metastases in MMTV-Cre:Rb^f/f:Pten^f/f versus MMTV-Cre:Rb^f/f but not versus MMTV-Cre:Pten^f/f mice (Fig. 5b). A few

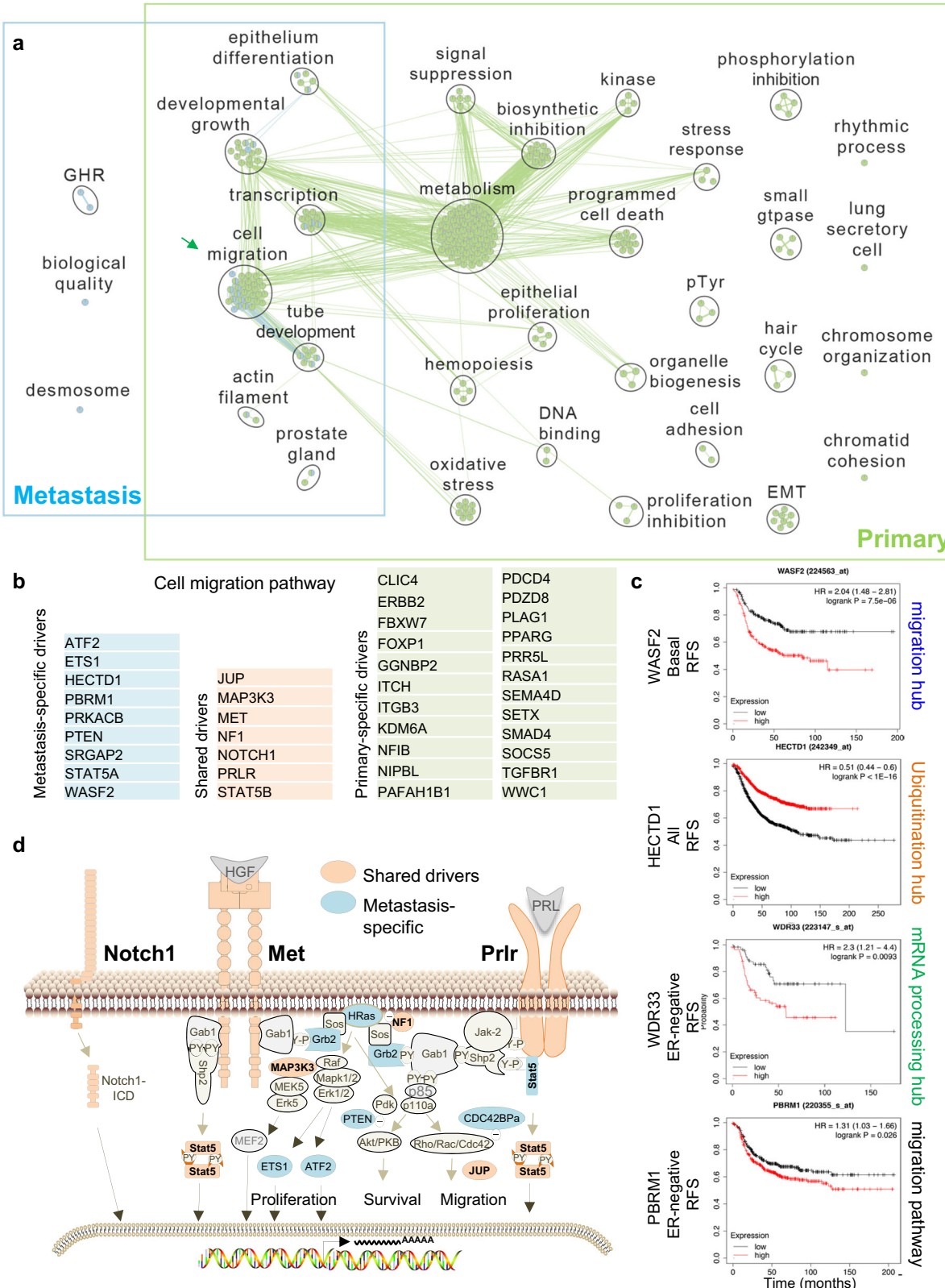

**Fig. 3 | g:Profiler uncovers cell migration as a common biological process in primary and metastatic gCISs comprising all 7 S-drivers. a** g:Profiler of gCISs in primary vs metastases using gene ontology (GO) for biological processes. Demarcated are overlapping pathways including 'cell migration' in both compartments. **b** Lists of the P- M- and S-gCISs in the 'cell migration' pathway. All 7 shared oncogenic gCISs (Fig. 1g) are included in the 'cell migration' pathway. **c** Kaplan–Meier relapse-free survival (RFS) curves of selected genes from the cell migration pathway, ubiquitination pathway and pre-mRNA processing hubs (Fig. 2e) as well as the metastasis-specific 'cell migration' pathway (**b**), using kmplot.com. RFS curves for other genes from these pathways are shown in subsequent figures and supplemental Fig. S2. **d** Schematic presentation of oncogenic pathways induced by gCISs on the MET, Prolactin receptor (PRLR) and NOTCH1 pathways, promoting cell proliferation, survival and migration through transcriptional, protein-protein interactions and post-translational modifications. S-gCIS are in orange; M-gCIS in light blue.

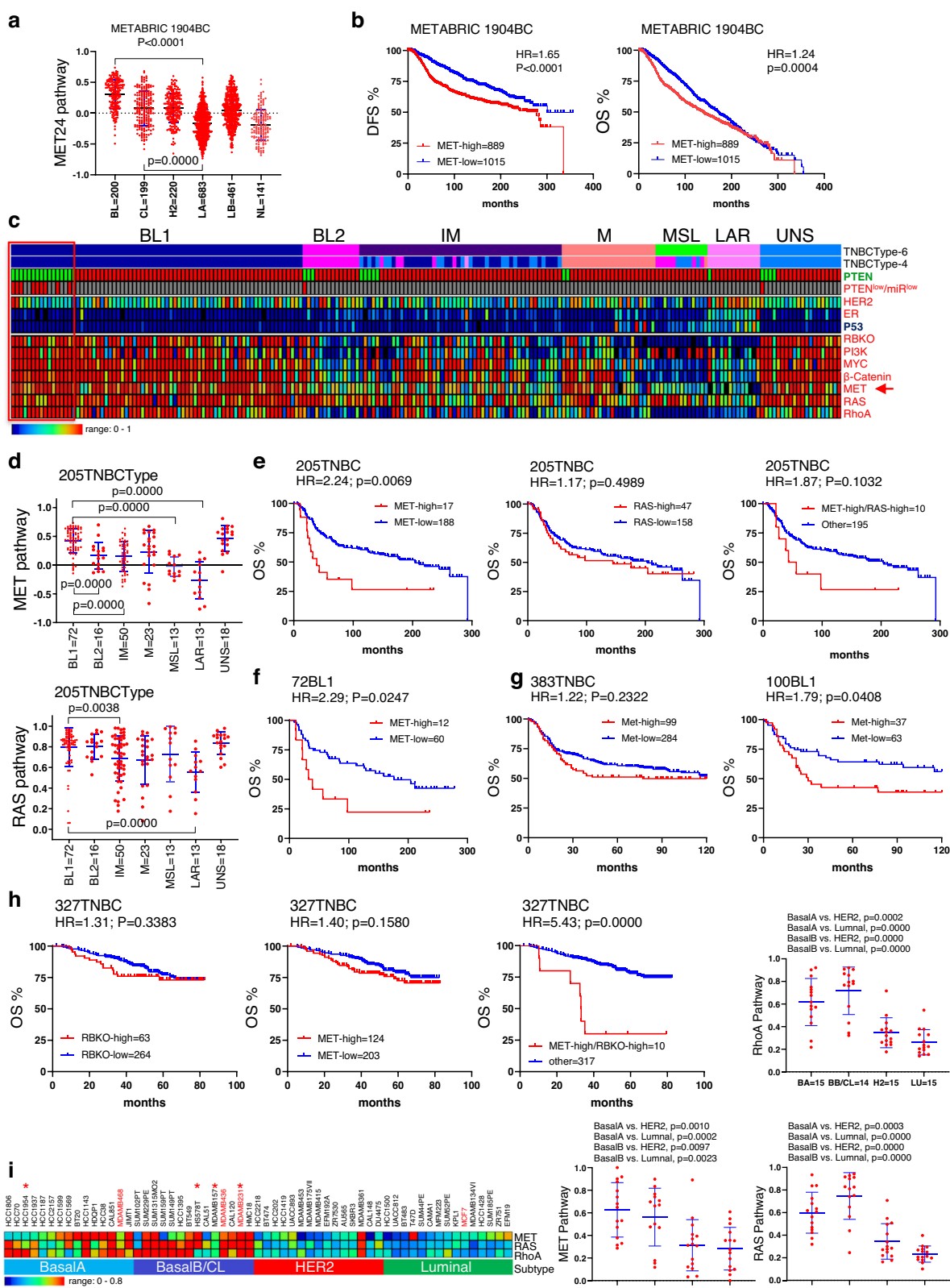

MMTV-Cre:Rb^f/f:Pten^f/f mice exhibited exceptionally high incidence of lung metastases, raising the question of whether they have acquired p53 mutation. However, immunostaining revealed that these lesions did not express high level of p53 (Supplementary Fig. S4b), suggesting they are unlikely driven by a stabilizing p53 mutation.

In keeping with this genetic analysis, we found that *RB*-pathway loss cooperates modestly with *PTEN* loss in increasing the hazard ratio

of breast cancer patients (Fig. 5c). Thus, using the EGAS00001001753 human breast cancer dataset[81], the disease specific survival of RBKO-high(i.e. RB-low)/PTEN-low breast cancer samples was significantly (*P* < 0.0001) lower than RB-high/PTEN-high with HR = 1.78 compared to RB-low vs RB-high (HR = 1.62) or PTEN-low vs PTEN-high (HR = 1.35). These tumor suppressors do combine to promote highly aggressive TNBC in cooperation with TP53 loss, and TNBC with mutations in all

**Fig. 4 | Impact of high cMET but not RAS pathway activity on clinical outcome of triple-negative breast cancer patients in cooperation with RB-loss.**
**a** Expression of a cMET signature, MET24, developed for hepatocellular carcinomas (HCC), in indicated molecular breast cancer subtypes, with highest level in basal-like breast cancer. In **a**, **d**, **i**, $P = 0.0000$ denotes $P < 0.0001$, using PRISM and two-tailed, unpaired $t$ test; error bars represent SD. **b** Kaplan–Meier disease-free (DFS) and overall (OS) survival curves for breast cancer patients segregated based on MET24 signature level. HR denotes hazard ratio. $P$ values in **b**, **e**, **h** calculated by Log-rank (Mantel–Cox) test. **c** cMET signature expression in 6 and 4 different triple-negative breast cancer (TNBC) subtype classification with highest expression in BL1 together with RBKO high, PI3K-, MYC-, WNT-, RAS- and RHOA-signaling high and PTEN and TP53 loss. MET24-high samples overlap with the most aggressive TNBC lesions (demarcated by the red box), defined by PTEN-loss and 5 miRNA-low (PTEN^low/miR^low, red) as described[70]. BL1 Basal-like 1, BL2 Basal-like 2, M

mesenchymal, MSL mesenchymal stem–like, IM immunomodulatory, LAR luminal androgen receptor, UNS unspecified. **d** Levels of MET24 and RAS signatures in the different TNBC subtypes. MET24 is significantly higher in BL1 versus other subtypes; RAS pathway activation is seen in BL1, BL2 and MSL. **e** Kaplan–Meier OS curves showing MET but not RAS pathway activity identifies TNBC with unfavorable prognosis. **f** Kaplan–Meier OS curve showing MET pathway-high identifies BL1 TNBC patients with exceedingly poor prognosis. **g** OS curve of an independent cohort showing MET pathway-high segregates BL1 but not all TNBC patients into relatively fair versus poor prognosis. **h** Effect of RB loss, MET signature high or both on OS of TNBC patients in the SCAN-B 327 TNBC cohort. Analysis of two additional cohorts is shown in supplementary Fig. S3. **i** Heat map and graphic presentation of MET24, RAS and RHOA signature levels in breast cancer cell lines classified by PAM50. Cell lines marked in red were used to characterize gCISs; those marked by asterisks were used to test the effect of RB depletion on cell migration.

three tumor suppressors are particularly lethal[36,82,83]. Together, this analysis suggests that genetic cooperation identified in SB screens involves more than two genes (*Rb*-loss plus two different gCISs). Indeed, both *Fbxw7* and *Pten* are disrupted together with other gCISs in addition to the *Rb*-deficient background (Fig. 1f; Supplementary Data S1). We therefore determined, in subsequent analysis, the effects of selected genes from our SB screens on growth and migration of established human breast cancer cell lines with known *RB* status, MET and RhoA signaling (Fig. 4i).

### Functional analysis of *SRGAP2* and *CDC42BPA*
Kaplan–Meier OS analysis of breast cancer patients revealed that low expression of *FBXW7, SRGAP2, MTMR3* as well as *CDC42*, the partner of *CDC42BPA*, is associated with poor prognosis (Supplementary Fig. S5a). In contrast, low expression of *CDC42BPA* marks breast cancer patients with better prognosis. Opposing correlations between *CDC42* and *CDC42BPA* expression and OS are also seen in basal-like breast cancer (Supplementary Fig. S5). However, Oncoprint analysis of *CDC42BPA* reveals deep deletions of the gene are almost exclusively found in metastatic but not primary breast cancer (Supplementary Fig. S5b), suggesting that loss of this gene may promote metastasis.

Western blot analysis revealed relatively variable expression of CDC42BPA, SRAGAP2 and MTMR3, and uniform expression of FBXW4 in several TNBC cell lines (Fig. 5d). To determine the impact of each gCIS on cell growth, isogenic lines were generated in which the corresponding genes were stably knocked down via lenti-shRNA, using commercially available constructs, alongside lenti-scrb (scrambled) control lines (Supplementary Fig. S6a).

To test whether these metastatic-specific gCISs modulate cell migration, we performed scratch-wound assays on these isogenic cells (Fig. 5e, Supplementary Fig. S7a–c). Knockdown of *FBXW7* and *CDC42BPA* but not *SRGAP2* or *MTMR3* promoted migration in the highly migratory BasalB/Claudin-low cell lines MDA-MB-436 and MDA-MB-231. In contrast, in the poorly migrating BasalA line MDA-MB-468, depletion of *CDC42BPA* had no significant effect whereas depletion of *FBXW7* slightly reduced cell migration. These results are in line with previous observations that depletion of *CDC42* diminishes migration in cancer cells with moderate metastatic potential but enhances cell migration in highly metastatic TNBC cell lines such as MDA-MB-231[84]. Indeed, depletion of *CDC42BPA* induced larger and rounded/elongated morphology in MDA-MB-231 and MDA-MB-436 cells but had no discernible effect on the shape of MDA-MB-468 cells (Supplementary Fig. S6b).

Knockdown of *CDC42BPA (MRCKA)* was previously shown to cooperate with Rho-kinase (ROCK) inhibition to suppress phosphorylation of Myosin Light Chain 2 (MLC2) on Thr18/Ser19, thus controlling contractility[85]. In both MDA-MB-231 and MDA-MB-468 TNBC cells, we found that while the ROCK inhibitor, Fasudil, significantly suppressed pThr18/Ser19-MLC2, *CDC42BPA* depletion

had no effect (Fig. 5f top; Supplementary Fig. S7d). However, *CDC42BPA* depletion diminished the inhibitory effect of Fasudil on pThr18/Ser19-MLC2. In accordance, Fasudil treatment slowed down the enhanced cell migration seen following *CDC42BPA* depletion (Fig. 5f bottom; Supplementary Fig. S7e). Thus, CDC42BPA modulates MLC2 phosphorylation and cell migration by antagonizing the effect of ROCK.

Cell viability was measured using trypan blue exclusion analysis and MTT assays. In MDA-MB-436, trypan blue assays revealed increased viability only upon *SRGAP2* depletion, whereas the MTT assays showed increased mitochondrial activity following *SRGAP2, CDC42BPA* and *FBXW7* depletion compared with control or MTMR3-silenced cells (Supplementary Fig. S7b). Significant increase in the MTT but not trypan blue assays was also observed following CDC42BPA depletion in MDA-MB-231 but not in MDA-MB-468 cells relative to control. These results suggested that loss of CDC42BPA function may promote mitochondrial activity. In agreement with this, total cellular ATP was elevated in *CDC42BPA*-deficient MDA-MB-436 and MDA-MB-231 but not in MDA-MB-468 cells relative to all other knocked-down or control cells (Supplementary Fig. S7c).

shRNA-mediated depletion of *RB* in diverse RB+ breast cancer lines also increased migration in many, though not all cell lines (Fig. 5f; Supplementary Fig. S8a). Specifically, RB knockdown increased cell migration in HCC1954, Hs578t and MDA-MB-231 cells but not in MDA-MB-157 cells, the latter of which exhibited the lowest levels of *MET*, *RAS* and *RhoA* signaling (Fig. 4i). In an independent series of experiments, we knocked-down *CDC42BPA*, *RB* or both genes in MDA-MB-231 cells, which normally express pRB (Supplementary Fig. S8b). Depletion of each gene alone significantly accelerated cell migration in wound scratch assays (Supplementary Fig. S8c). Combined deletion of both *CDC42BPA* and *RB* did not further enhance migration, perhaps because these genes operate on the same pathway and/or that loss of each gene alone saturates the migration potential of these cells.

We next probed the effect of stable depletion of these genes on xenograft growth and lung metastasis of MDA-MB-436 cells, orthotopically injected into immune-deficient NSG mice. Two million tumor cells from each isogenic line were transplanted into #4 mammary glands of recipient female mice (≥6 mice per group), and tumor weight and lung metastasis were determined at the same end point. *FBXW7*-depletion used as control increased primary tumor weight to near significant level ($P = 0.062$; Fig. 5h); *CDC42BPA* depletion significantly increased both primary tumor weight ($P = 0.0009$) and lung metastases ($P = 0.045$; Fig. 5i; Supplementary Fig. S7f); whereas *MTMR3* knockdown mildly but not significantly increased tumor growth ($P = 0.26$), but robustly induced lung metastases ($P = 0.0007$; Fig. 5j).

### Functional analysis of *FBXW4* and *WDR33*
We next tested the effect of selected genes from the two other hubs identified in our metastasis-specific SB screens, *FBXW4* and *WDR33*, on

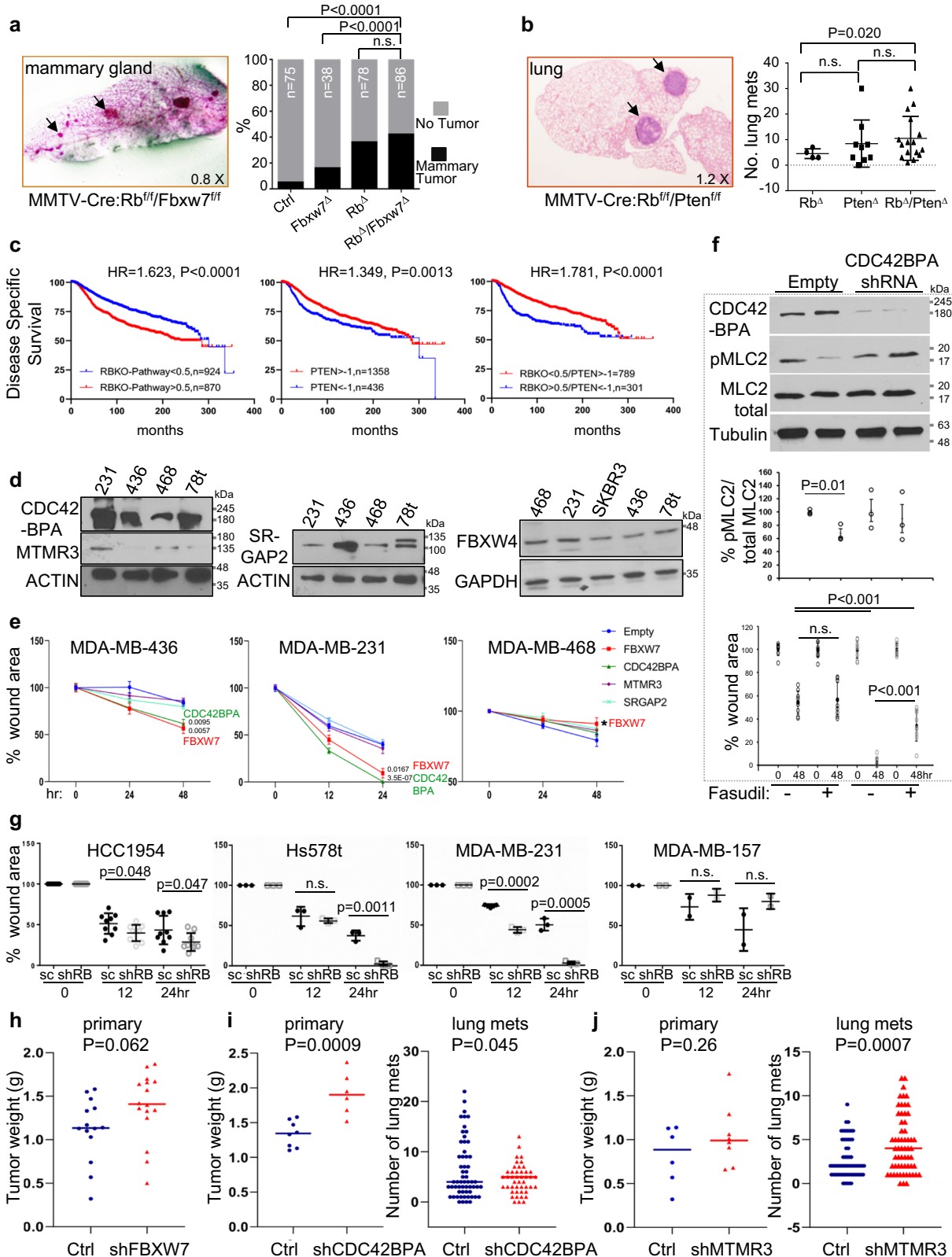

cell growth, migration and tumorigenesis. Over-expression of *FBXW4* (F-Box and WD Repeat Domain Containing 4) via a recombinant adenovirus vector suppressed cell proliferation (MTT assay), increased cellular senescence (β-galactosidase assay), and reduced cell migration (scratch-wound assay) in MCF7 luminal and MDA-MB-231 TNBC cells (Fig. 6a–c). Conversely, lenti-shRNA-mediated knockdown of *FBXW4* increased cell proliferation in MDA-MB-231 cells, and promoted MCF7 cell migration as

well as tumor growth following orthotopic transplantation into NSG mice (Fig. 6d–f). Consistent with these results, low *FBXW4* RNA expression correlates with poor clinical outcome in breast cancer patients with HR = 2.19 (*P* = 0.0001; Fig. 6g). As a prelude to the identification of cellular targets for FBXW4, we analyzed the effect of its depletion on multiple factors involved in cell proliferation and survival (BCL2, BAX, p27, P21, MYC, ATG3, PUMA ATM, Cyclin-B, NANOG, MRPL37, E-CADHERIN,

**Fig. 5 | Effect of selected genes targeted by gCISs on tumorigenesis and cell migration. a** Incidence of microscopic mammary tumors in MMTV-Cre:Rb$^{f/f}$:Fbxw7$^{f/f}$ female mice relative to single mutant mice. Left, whole mount stained mammary gland from a representative MMTV-Cre:Rb$^{f/f}$:Fbxw7$^{f/f}$ mouse with multiple microscopic tumors (arrows). Right, quantification of mammary tumors in indicated mice. *P* value calculated by one-way ANOVA, Tukey's multiple comparison test. **b** Incidence of microscopic lung lesions in MMTV-Cre:Rb$^{f/f}$:Pten$^{f/f}$ mice relative to single mutant mice. Left, cross section through a representative lung; arrow points to a large lung lesion. Right, number of lung mets in indicated mouse strains. Error bars represent SD; *P* value calculated by one-way ANOVA, Tukey's multiple comparison test. **c** Kaplan–Meier Disease-free survival curves for breast cancer patients with high RBKO pathway (loss; left), low PTEN mRNA level (center), or RBKO high (loss) plus PTEN mRNA low (right) compared to all other genotypes. HR hazard ratio. *P* values calculated by Log-rank test. **d** Western blot analysis for expression of indicated M-targets in different TNBC cell lines: MDA-MB-231; MDA-MB-436; MDA-MB-468; Hs578t; and HER2-enriched SKBR3. **e** Effect of shRNA-mediated *FBXW7, CDC42BPA, SRGAP2* or *MTMR3* depletion on cell migration (scratch-wound assays) in indicated TNBC lines. *P* values calculated by unpaired, two-tailed student *t*-test. * denotes

*P* = 0.0313 by one-sided *t* test. **f** CDC42BPA depletion counteracts the effect of Rho-kinase inhibitor, fasudil, on pMLC2 phosphorylation and cell migration. Top, MDA-MB-231 cells stably transduced with empty or *CDC42BPA* lenti-shRNA virus were treated with fasudil (25uM) or vehicle alone followed by western blots for CDC42BPA, anti-pMLC2-Thr18/Ser19 or total MLC2 with tubulin as loading control. Middle, statistical analysis on three independent biological replicates (see supplemental Fig. S7d). Bottom, statistical analysis on 9 independent scratch assays (see supplemental Fig. S7e). *P* values by unpaired two-tailed student *t*-tests. **g** Effect of RB depletion on TNBC cell migration in indicated cell lines. Error bars represent SD. **h** Effect of *FBXW7* depletion on primary tumor formation of MDA-MB-436 cells following orthotopic transplantation into immune-deficient NSG mice (≥6 mice per group). *P* values calculated by one-way ANOVA, Tukey's multiple comparison test. **i, j.** Effects of *CDC42BPA* or *MTMR3* depletion on primary tumor formation and metastases of MDA-MB-436 cells following orthotopic transplantation into NSG mice (≥6 mice per group). Shown are primary tumor weights at end point (left) and number of lung mets (right) from multiple lung sections. *P* values calculated by one-way ANOVA, Tukey's multiple comparison test. Source data are provided as a Source data file.

VINCULIN). Transient or stable depletion of FBXW4 consistently increased levels of the BCL-2 survival factor, whereas over-expression of FBXW4 reduced BCL-2 levels (Fig. 6h). In addition, *FBXW4* depletion induced expression of the Mitochondrial Ribosomal Protein L37 (MRPL37), a transcriptional target of pRb-E2F[36] (Fig. 7i).

As noted, high expression of the pre-mRNA 3′ end processing protein WDR33 correlates with poor clinical outcome (Fig. 3c). Western blot analysis WDR33 uncovered multiple forms and varied expression in different TNBC lines; it was low in MDA-MB-23 but high in MDA-MB-468 cells (Fig. 6j–k). Stable over-expression of WDR33 via a lentivirus in MDA-MB-231 cells significantly increased cell proliferation. Conversely, lenti-shRNA mediated depletion of WDR33 in MDA-MB-468 cells significantly reduced cell proliferation, though it had no effect on cell migration. Together, these results demonstrate that metastasis-specific components from the three hubs identified in our SB screens as well as *RB*, control proliferation, motility and tumorigenesis in a cell/context specific manner.

## Shared and compartment-enriched oncogenic pathways in breast cancer

The aforementioned SB mouse analysis suggests a model in which S-drivers cooperate with distinct compartment-specific oncogenic networks to promote local versus metastatic disease. To determine whether similar mechanisms operate in human breast cancer, we assessed the activity of 25 different signaling pathways/oncogenic signatures[86,87] as well as MET24 in a test cohort of 97 primary/120 unpaired mets (GSE81954/GSE56493), and 83 paired primary and metastatic tumors (GSE147322)[88,89]. Pathway activity was calculated for all breast cancer samples or for specific PAM50 subtypes (basal-like, HER2-enriched, luminal A and luminal B), segregated into primary lesions and metastases (Fig. 7a; Supplementary Fig. S9a). Differences in mean pathway activity between primary and metastasis is shown as Δmean and significance was calculated by 2-tailed student *t*-test.

When all tumors were considered, RhoA and PI3K pathways were significantly (*P* < 0.0001) and robustly (ΔMean > 0.27) elevated in metastases versus primary tumors in both the test and validation cohorts, and may therefore represent metastatic-enriched oncogenic drivers (Fig. 7a; Supplementary Fig. S9b). Conversely, the TGFβ, EGFR and STAT3 pathways were significantly elevated in primary versus metastases in both the test and validation cohorts. This pattern is consistent with the tumor-suppressor effect of TGFβ on primary tumor formation and its subsequent conversion to a promoter of metastatic growth[90–92], and the paradoxical effect of EGFR during breast cancer progression[93]. STAT3 is considered a major driver of metastatic breast cancer[94,95], yet, components of this pathway exhibit loss-of-function

mutations in metastatic breast cancer[96,97], in line with our observed decrease in STAT3 pathway activity in metastases. Two additional smaller cohorts with 36 (GSE57968) and 45 (GSE184869) paired primary/met samples[98,99] further corroborated these results, showing robust enrichment of RhoA and PI3K signaling in metastases, and significant enrichment of TGFβ, EGFR and STAT3 pathway activity in primary tumors (Supplementary Fig. S9c). The exception was the PI3K pathway, which was robustly but not significantly induced in the small 36 paired cohort. Multiple other pathways such as MET, RB-loss, RAS, TP53, β-catenin and RBKO, exhibited no statistical significance between primary versus metastatic lesions, and may thus be considered as S-drivers. A quantification of combined data of all four cohorts for M-enriched (RhoA and PI3K), S- (RAS, RB-loss, MET) and M-enriched (STAT3, EGFR, TGFβ) pathways is shown in Fig. 7b.

Similar analysis of each breast cancer subtype was more challenging as the number of patients, especially in the basal-like subtype, was much reduced in the test cohort and even smaller in the validation cohort. Indeed, in basal-like tumors, while the RhoA, SRC, AKT, E2F4KO and PI3K pathways were significantly and highly elevated, and TGFβ and EGFR pathways were highly reduced in metastases vs primary lesions in the test cohort, these effects could not be validated in the independent cohort (Fig. 7c; Supplementary Fig. S9b). On the other hand, in the HER2-enriched subtype, Rho signaling was significantly, robustly and reproducibly induced in metastases vs primary lesions, whereas in Luminal A and Luminal B, induction of both RhoA and PI3K signaling was observed in both the test and validation cohorts. In addition, in luminal A breast cancer, TGFβ signaling was significantly reduced in metastases relative to primary tumors. Additional pathways, marked in bold, in particularly SRC, were induced, while EGFR and STAT3 pathways were reduced in metastasis vs primary lesions in the test cohort in all or most subtypes, respectively. Finally, most other pathways noted above including MET, RAS and RBKO were equally altered in both compartments in the different breast cancer subtypes, or, in the case of ER signaling, elevated in both primary and metastatic luminal tumors, hence representing S-drivers. Thus, for these 26 pathways, S-drivers are subtype-specific, whereas P- and M-drivers appear similar across subtypes.

## TNBC patients with S- (RB-loss) plus M- (RHO-high) driver pathways exhibit poor prognosis

To investigate the cooperative impact of S- and M-driver pathways on clinical outcome, we determined the OS of TNBC patients with high S- (RBKO, MET) and M- (RhoA signaling) driver pathways in three independent clinical cohorts (METABRIC 205 TNBC; SCAN-B 327 TNBC; FUSCC 360 TNBC). TNBC patients with high RBKO (RB-loss) plus high RhoA signaling exhibited worse clinical outcome compared

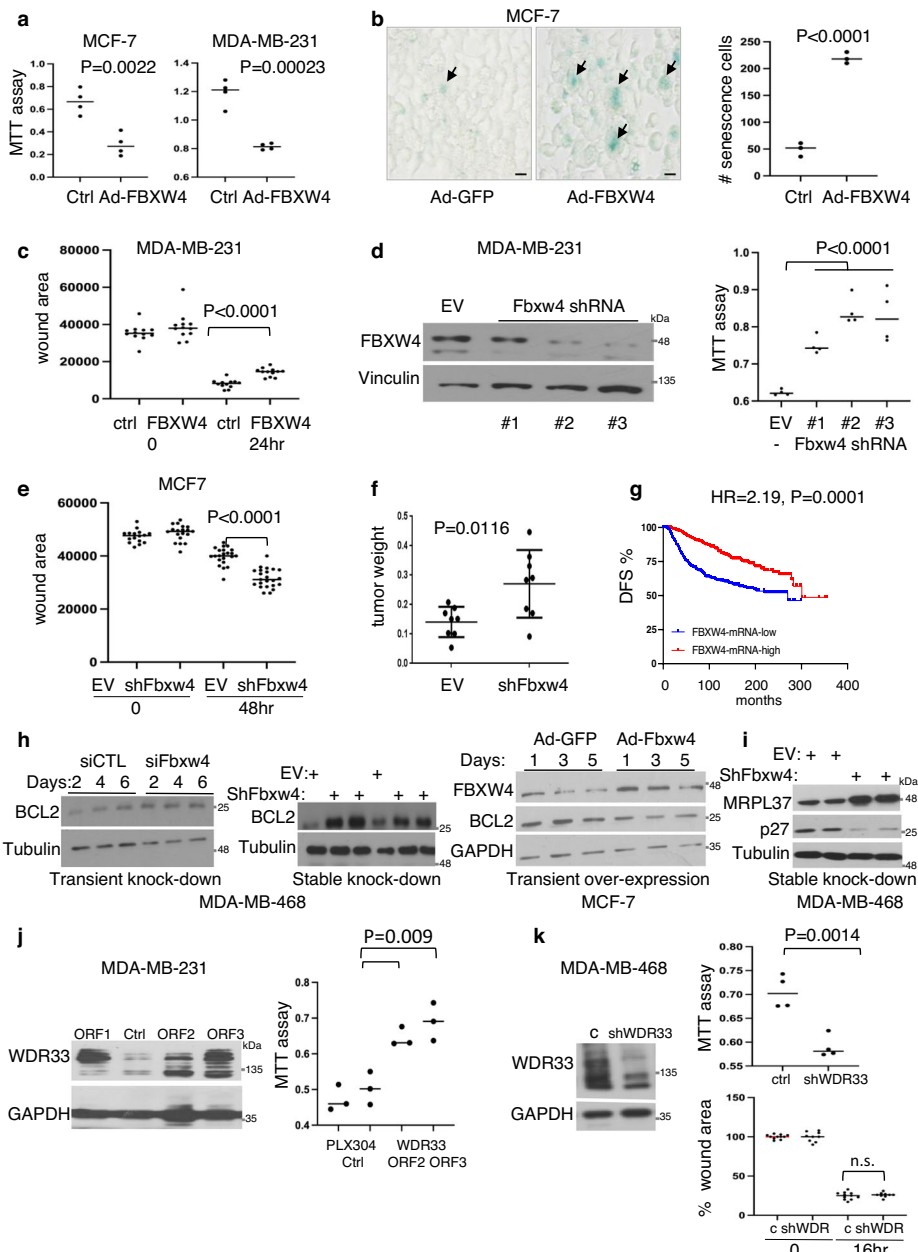

**Fig. 6 | Effects of *FBXW4* and *WDR33* on breast cancer cell growth and tumorigenesis. a** Growth suppression by adenovirus-FBXW4 in MCF7 luminal and MDA-MB-231 TNBC cells, determined by MTT ([3-(4,5-Dimethylthiazol-2-yl)−2,5-Diphenyltetrazolium Bromide]) assays. All *P* values were determined by two-tailed, unpaired t test; error bars represent SD. **b** Induction of senescence (senescence-associated β-galactosidase assays; blue; arrows) by adenovirus-FBXW4 in MCF7 cells. Scale bar, 50 μm. Right, quantification of results from a representative experiment. **c** Suppression of cell migration (scratch-wound assays) by adenovirus-FBXW4 in MDA-MB-231 TNBC cells. **d** Left, western blot analysis demonstrating efficient depletion of FBXW4 via lenti-shRNA (clone #3) versus control, Empty Vector (EV). Right, effect of FBXW4 depletion via shRNA clones on cell proliferation (MTT assays). * denotes *P* < 0.05 by two-tailed student *t*-test (*n* = 3). **e** Induction of cell migration following FBXW4 depletion in MCF7 cells. **f** Significant increase in tumor formation following orthotopic transplantation of FBXW4 depleted MCF7 cells into NSG mice (8 mice per group) versus control, EV-transduced MCF7 cells.

**g** Kaplan–Meier disease-free survival (DFS) curve of breast cancer patients expressing low vs high *FBXW4* mRNA levels. *P* value by Log-rank (Mantel–Cox) test. **h** Western blots showing that transient (left) or stable (center) depletion of FBXW4 increases BCL2 expression whereas FBXW4 over-expression decreases BCL2 levels in indicated cells. Representative blots of 3 biological experiments each. **i** Western blots showing that stable knockdown of FBXW4 increases MRPL37 and suppresses p27[KIP1] expression. Representative blots of 3 biological experiments. **j** Left, generation of MDA-MB-231 TNBC cell lines over-expressing WDR33 via recombinant lentivirus. Right, induction of cell proliferation following WDR33 over-expression as determined by MTT assays. Representative experiment of 3 biological experiments each performed in triplicates. **k** Left, generation of a WDR33-knocked-down MDA-MB-468 TNBC cells. Right, top, WDR33 depletion reduced cell proliferation by MTT assays (right, top) but had no effect on cell migration in wound scratch assays (Right, bottom). Representative experiment of 3 biological experiments each performed in triplicates. Source data are provided as a Source data file.

to patients with only RB-loss or only high RhoA signaling in the three cohorts with HRs of 1.94 (*P* = 0.0097), 2.2 (*P* = 0.022); and 5.17 (*P* = 0.0004), respectively (Fig. 8a, Supplementary Fig. S3). For the METABRIC cohort, the median OS for Rb-loss/RhoA-high was

~67 months (5.58 years) compared to 200 months (16.67 years) for all other tumors, i.e. over 10 years difference. Such consistent cooperation was not observed between MET, β-catenin/WNT or PI3K pathways and Rho signaling (Supplementary Fig. S3). The prognosis of TNBC

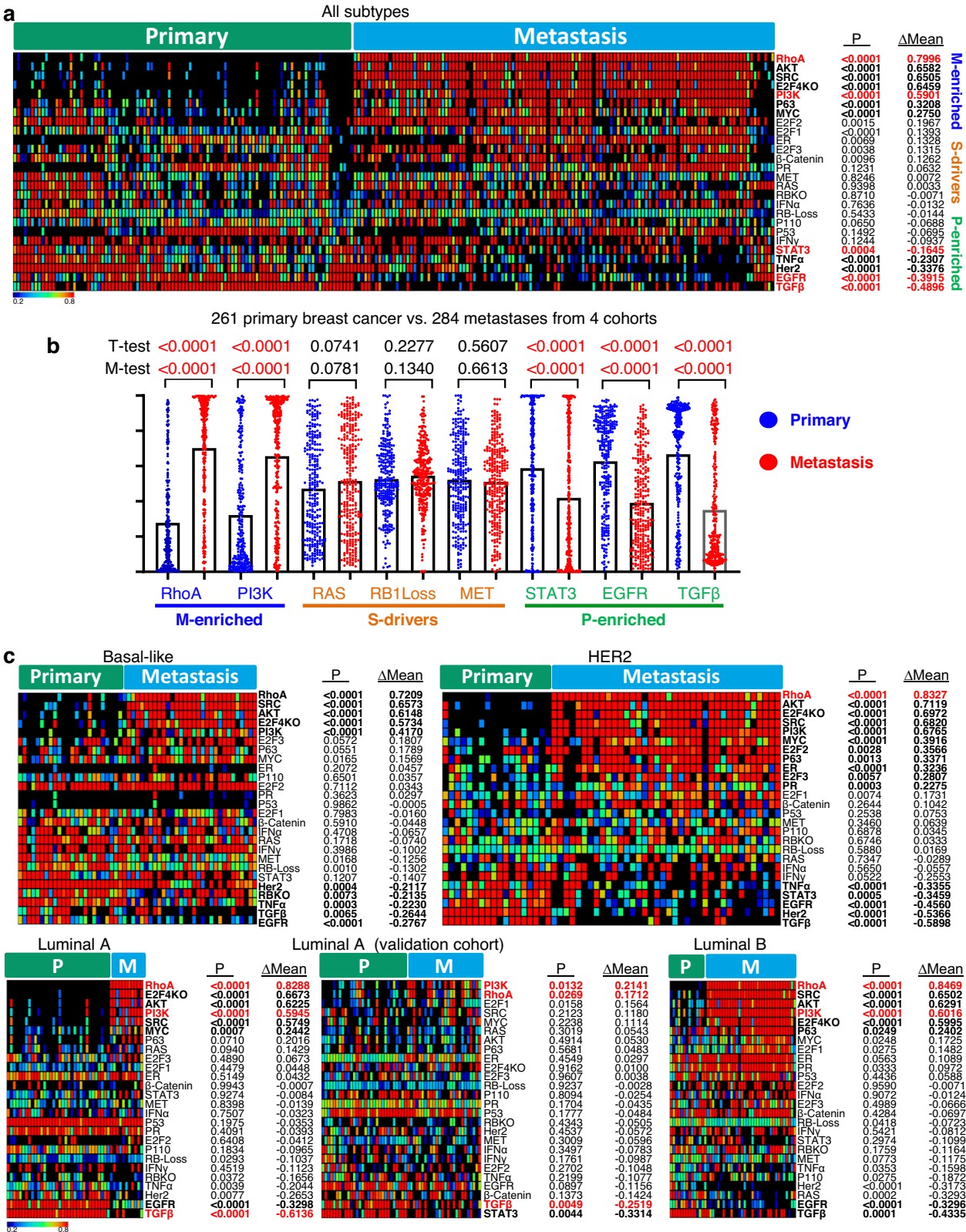

patients with RB-loss and high MET signaling was comparable to or even worse than that seen in RB-loss/high RhoA signaling in these 3 cohorts, with HRs of 2.67 ($P = 0.009$), 5.43 ($P = 0.0001$), and 3.63 ($P = 0.0081$), respectively (Fig. 4h, Supplementary Fig. S3). Thus, TNBC patients with RBKO-MET (two S-drivers) or RBKO-RhoA (S-driver plus M-driver) pathway activation exhibit extremely poor clinical outcome and should be prioritized for therapy.

## Pharmacological inhibition of S- (RB-loss; MET-high) plus M- (RHO-high) driver pathways

Oncogenic alterations in components of Rho GTPases involved in cell motility/migration were identified in primary breast cancer[100–102]. To explore whether oncogenic alterations in Rho/Rac1/CDC42 signaling are further enriched in metastatic vs primary breast cancer, we compared the rate of missense mutations and copy number alterations

**Fig. 7 | Primary-, metastatic- and shared-oncogenic pathways in human breast cancer. a** Heat map for activity of 26 pathways in primary vs metastatic breast cancer of all patients in the test cohort. In bold are pathways that are significantly and robustly different between primary lesions and metastases. Highlighted in red are pathways that are also altered in the validation and two additional cohorts (supplementary Fig. S9b, c). *P* values represent difference between primary and metastases by unpaired two-tailed *t*-test; ΔMean, difference in mean in the two compartments. *P* = 0.0000 denotes *P* < 0.0001. Differences in pathway activity

were calculated; color scale denotes pathway activity. **b** Jitter plots showing combined results from four different cohorts for M-enriched (RhoA; PI3K), S- (RAS, RB-loss, MET) and P-enriched (STAT3, EGFR, TGFβ) pathways. Significance was calculated by student *T*-test and Mann–Whitney test (M). *P* values calculated by unpaired two-tailed *t*-test. **c** Heat map for activity of the 26 pathways in paired primary vs metastatic basal-like, HER2, Luminal A and Luminal B. For Luminal A, the validation cohort is also shown. Color scale denotes pathway activity. Additional analysis of the validation cohort is shown in supplementary Fig. S9b.

(CNA) in 42 genes associated with these pathways (listed in Supplemental Fig. S10a). We used the Memorial Sloan Kettering Cancer Center: MSKCC2018 dataset with paired primary and metastatic breast cancer patients[103] and MSKCC2022[104] from the same cohort with a longer follow-up of metastatic lesions. Overlapping samples between the two datasets were removed to obtain unique Sample_ID and Patient_ID. Of the 42 genes encoding Rho/Rac1/CDC42 associated proteins, sequencing data of only 7 (*MAP3K1, PAK1, PDGFRA, PIK3CA, PIK3CG, RAC1* and *RHOA*) were available, and of these, PDGFRA and RHOA showed significant enrichment of missense mutations in metastatic vs primary tumors (Supplementary Fig. S10b). However, these mutations affected only a small percentage (<2%) of metastases. Notably, despite the increase in PI3K pathway activity (Fig. 7), and the frequent mutations in PIK3CA in breast cancer, there was no further increase in the PIK3CA mutation rate in metastases vs primary lesions. Furthermore, no significant changes were observed in the landscape of PIK3CA hot-spot mutations in the two compartments (Supplementary Fig. S10c). CNA data were available for 10 of the 42 Rho/Rac1/CDC42 signaling genes (*PAK1, PIK3CA, CDC42, MAP3K1, PDGFRA, PIK3CB, PIK3CG, PIK3R1, RAC1, RHOA*), and of these, only *PAK1* (P21/Cdc42/Rac1-Activated Kinase 1) showed a significant increase in copy number gain in metastasis (~11–12%) vs primary (~7.3–7.5%) samples and patients (*P* < 0.01; Supplementary Fig. S10d). Copy number gain of *PAK1* alone did not correlate with a significant worse prognosis (HR = 1.2; *P* = 0.11; Supplementary Fig. S10e). However, high PAK1 mRNA expression correlated with poor overall survival in two independent OS cohorts, with HRs of 1.88 (*P* = 0.094; SCAN-B) and 2.5 (*P* = 0.0029; METABRIC; Supplementary Fig. S10f), but not in a progression-free survival cohort (TCGA), suggesting that high PAK1 induction at the mRNA level is required to achieve tumorigenic impact. While no new mutations in RHO or PI3K signaling could account for the increase in their signaling activity in metastatic breast cancer, primary tumors with high levels of these pathways may be selected for metastasis. Alternatively, induction of RHO and PI3K signaling in metastatic breast cancer could be caused by additional genetic and epigenetic alterations, post-translational modifications, or external cues from the tumor microenvironment. Regardless of the specific upstream oncogenic drivers, inhibition of these pathways may suppress metastasis.

The Actin related protein 2/3 complex (Arp2/3) is induced by WASP in response to a CDC42-dependent signal to promote actin assembly required for lamellipodia extension and directional cell migration[105,106], and can be specifically inhibited by CK666[107]. Effective inhibitors of the S-driver, MET, include Tivantinib[108] and SGX523[109], whereas the S-driver, RB-pathway-loss, can be successfully inhibited in pre-clinical models of TNBC using the WEE1 kinase inhibitor, MK1775 (Adavosertib, AZD1775)[83] or the Aurora A kinase inhibitor, Alisertib[110]. Determining the impact of these drug combinations on tumor cell survival and migration is important to demonstrate cooperation of S- and M-enriched oncogenic drivers, and a critical first step in assessing S-driver/S-driver- vs S-driver/M-driver-based therapies. To this end, we first analyzed single and combinations of these drugs using MTT and IncuCyte proliferation assays (Fig. 8b; Supplementary Fig. 11a–e). MK1775 (Fig. 8b) or CK666 (Supplementary Fig. 11c) plus Tivantinib had particularly strong inhibitory effects on cell proliferation/survival relative to other combinations.

Next, we used an IncuCyte-based scratch-assay to quantify the impact of these drug combinations on relative wound density (RWD). The assay was done in the absence of cell cycle inhibitors, hence the effects of these drugs could be assessed on both cell migration and survival in a single assay. For inhibitors of RB-loss (MK1775; Alisertib) and MET (Tivantinib; SGX523), we used $IC_{50}$ concentrations, whereas for the Arp2/3 inhibitor (CK666), we chose a drug concentration that had no effect on growth/viability based on the cell proliferation assays (Supplementary Fig. 11f). Combinations of the RB-loss inhibitors (MK1775; Alisertib) with the Arp2/3 inhibitor, CK666, were even more potent than combinations of RB-loss plus MET inhibitors (Tivantinib; SGX523; Fig. 8c; Supplementary Fig. 11g). Similar results were obtained with two additional TNBC cell lines (MDA-MB-231 and BT549), pointing to potential therapeutic avenues to prevent metastatic spread using combination treatments with inhibitors to two S-drivers, or S- plus M-drivers. Together, these results support a model in which S-drivers cooperate with compartment-specific P- and M-drivers to promote primary versus metastatic breast cancer with direct implications for progression, prognosis and prevention of metastatic disease (Fig. 8d)

## Discussion

In this communication, we describe Sleeping Beauty (SB) mutagenesis screens in the mammary gland conducted on both primary tumors and lung metastases. We identified oncogenic networks that drive metastatic mammary tumors in cooperation with loss of the tumor-suppressor Rb, and demonstrated the existence of primary (P)-specific, metastasis (M)-specific and shared (S) gCIS, the latter of which promote both primary and metastasis. The metastasis-specific gCIS were observed in drug naive animals, suggesting that oncogenic alterations found in metastatic samples may not only be due to drug selection but also represent genuine metastasis-promoting drivers. The M-gCIS form specific interactomes and pathways, the components of which correlate with poor clinical outcome in human breast cancer or were previously shown to promote cancer progression in other types of malignancies. TNBC patients with RB loss/MET-high (two S-drivers) or RB loss/RhoA high (S-driver plus M- enriched) exhibit exceedingly poor prognosis and may be prioritized for therapy. Thus, the S- and M-gCIS identified herein provide a rich resource for future basic and translational analysis. Importantly, we provide evidence that human breast cancer (4 different datasets) exhibits a similar organization with P-enriched, M-enriched and S-oncogenic drivers, and have demonstrated that drug combinations targeting S-drivers (RB-loss, MET) plus M-enriched (Rho signaling) cooperate to effectively block cell survival and migration of TNBC cells. Our analysis supports the idea that S-drivers cooperate with P-enriched or M-enriched drivers to promote local versus distal growth, respectively, and that targeting S- and M- but not P-enriched derivers may offer an effective modality for the prevention of metastatic breast cancer at the time of diagnosis (Fig. 8d).

Our model is supported by the following observations: (1) transposon-mediated mutagenesis in mice revealed overlapping/S-drivers as well as compartment-specific/enriched P- and M-gCIS. Key to this analysis was our ability to demonstrate, based on transposon-integration site analysis, the clonal relationship between primary and metastatic lesions; (2) string- and pathway analyses uncovered S- and

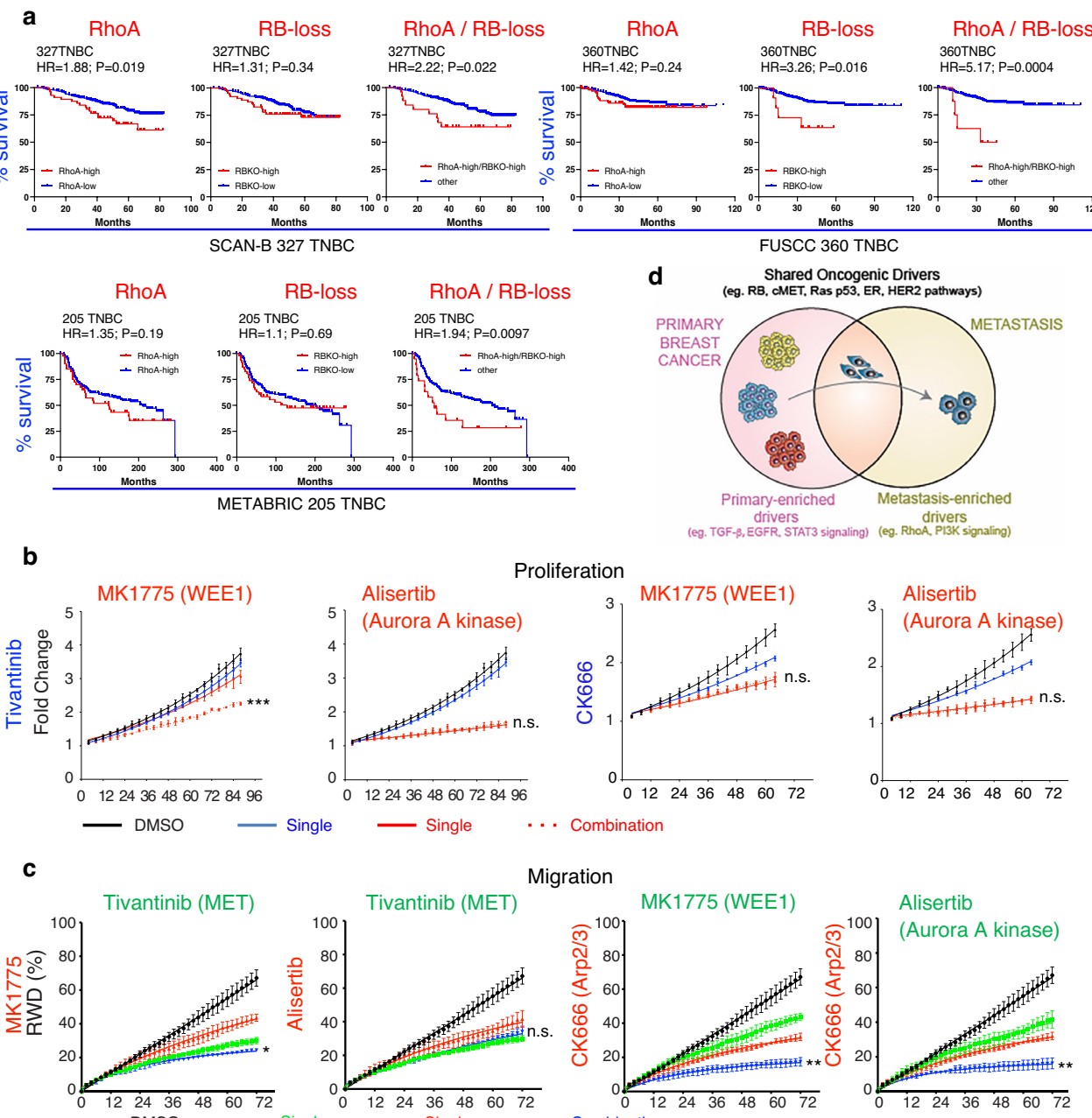

**Fig. 8 | Impact of RB-loss plus high RHO-signaling on prognosis, and effect of pharmacological inhibition of shared (RB-loss; MET-high) plus metastasis-enriched (RHO-high) oncogenic pathways on proliferation and migration of TNBC cells. a** Kaplan–Meier survival curves of TNBC patients expressing high RhoA signaling, RBKO high (loss) or both in three different clinical cohorts: SCAN-B 327 TNBC, FUSCC 360TNBC, and METABRIC 205TNBC. The entire analysis including other genes is shown in supplementary Fig. S3. **b** Representative IncuCyte proliferation assays comparing the effect of indicated inhibitors of RB-deficient cells (WEE1 kinase inhibitor, MK1775, 0.125 μM; Aurora A kinase inhibitor, Alisertib, 1.25 μM), MET (Tivantinib, 0.5 μM) and RHO signaling (Arp2/3 inhibitor, CK666, 120uM) alone and in combinations (additional analysis shown in supplementary Fig. S11). *P* values calculated by Welch's *T*-test: ***P < 0.0001 (MK1775 vs MK1775/Tivantinib). Error bars in **b**, **c** represent SD. **c** Representative IncuCyte migration assays over 72 h comparing the effect of indicated inhibitors of RB-loss (WEE1 kinase inhibitor, MK1775, 0.2 μM; Aurora A kinase inhibitor, Alisertib, 0.3 μM), MET (Tivantinib, 1 μM) and RHO

signaling (Arp2/3 inhibitor, CK666, 100 μM) alone and in combinations (*n* = 4 biological experiments each performed in triplicates; additional analysis shown in supplementary Fig. S11). *P* values calculated by Welch's *T*-test: *P = 0.027 (Tivantinib vs MK1775/Tivantinib); **P = 0.0017 (CK666 vs CK666/MK1775); **P = 0.0031 (CK666 vs CK666/Alisertib). Source data are provided as a Source data file. **d** A model depicting the oncogenic relationship between primary and metastatic breast cancer and impact on cancer progression and prevention. Subtype-specific shared-oncogenic drivers (S-divers) promote both primary and metastatic breast cancer (e.g. RB loss, p53-loss, MET, ER and HER2 gain/over-expression), and cooperate with primary (P-) enriched drivers (TGFβ, EGFR, STAT3 signaling) to promote primary breast cancer or with metastasis (M-) enriched drivers (RhoA, PI3K signaling) to induce metastatic disease. The S-drivers are subtype specific—e.g. ER pathway is elevated in both luminal P- and M-lesions, but not other subtypes such as TNBC. Combination therapy against two subtype-specific S-drivers, or an S-driver plus RHOA or PI3K signaling (M-enriched) may efficiently prevent metastatic dissemination.

site-specific hubs and pathways, highlighting distinct biological pathways/mechanisms in each compartment; (3) expression of multiple genes from the S- and compartment-specific hubs correlated with prognosis in human breast cancer, and many of these genes have been implicated in cancer progression in other types of cancer, underscoring their clinical importance; (4) functional analysis of representative hub genes demonstrated their effects on cell proliferation, migration and/or metastasis in xenotransplantation models; (5) S- and compartment-enriched pathway alterations were also identified in human breast cancer and specific subtypes; (6) aberrant expression of S- plus M- enriched pathways correlated with poor clinical outcome; (7) inhibitors of S- plus M-enriched pathways cooperated to restrict tumor cell survival and migration. As noted, our designation of P- and M-specific gCIS is based on filtered_clonal analysis. When using a less stringent cut-off at the filtered_subclonal level, as used in our clonal relationship between mammary tumors and lung metastases, we detected low levels of some gCIS across compartments, rendering them P- and M-enriched gCIS, which is in line with the human data. Thus, either de novo or pre-existing oncogenic alterations in primary lesions are enriched in and drive metastatic breast cancer.

The S-gCIS identified in our screens: Met, Prlr, Nf1, Jup, Map3k3, Stat5b and Notch1 as well as Rb-loss, are closely interconnected to form a hub that is well established to drive MET-RAS signaling, neoplastic transformation and cell motility/migration. The M-specific gCIS included three additional hubs involved in cell migration (e.g. CDC42BPA), protein ubiquitination (e.g. FBXW4) and pre-mRNA processing/splicing (e.g. WDR33). We confirmed that representatives of these hubs promote hallmarks of metastasis such as cell migration when over-expressed or knocked-down, and that these effects varied in different TNBC cell lines, thus demonstrating their genuine oncogenic impact in a context/tumor-specific manner. We also showed that expression of these hub-components correlates with poor clinical outcome of breast cancer patients. This includes Rho GTPase SRGAP2[48], CDC42BPA[49] and WASF2 (WAVE2)[50], ubiquitination enzymes SPOP[52], HECTD1[53], UBE2D3[54] and UBXN7, as well as mRNA processing and splicing factors CDC5L[56], PRPF6[57], PSPC1[58] and CWC22, implicated in progression of other types of cancer. The exact role of all these factors and their therapeutic potential in metastatic breast cancer warrant further, in depth investigation.

We next asked whether the pattern of P-specific, M-specific and S-gCIS observed in the SB mutagenesis screen also occurs in primary human breast cancer and metastases. Using pathway analysis for 26 different oncogenic drivers, we found a similar pattern of P-enriched, M-enriched and S-oncogenic pathways in four independent clinical cohorts of primary and metastasis human breast cancer. Specifically, consistent with their opposing effects on tumor initiation and progression[90–93], TGFβ and EGFR signaling were significantly elevated in primary tumors but diminished in metastases. We also observed reduced STAT3 signaling in metastases vs primary breast cancer samples in all four clinical cohorts, which concurs with reported inactivating JAK2 and STAT3 mutations in metastatic/relapse samples that are not observed in primary lesions[96,97]. At the other end of the spectrum, we found RHOA and PI3K pathways are elevated in metastatic versus primary lesions, thus representing metastasis-enriched oncogenic pathways. Other pathways such as MET, RAS, ER, TP53-loss and RB-loss are highly expressed in both primary tumors and metastases of all or specific breast cancer subtypes, and therefore represent shared oncogenic drivers. Notably, while the S-drivers are subtype-specific—e.g. ER is elevated in luminal whereas TP53 is lost in TNBC in both primary and metastasis—the P- and M-enriched pathways were found in all subtypes. Whether other P- or M-enriched pathways such as ubiquitination and mRNA processing pathways, observed in the M-specific hubs in our SB screens, are also enriched in all subtypes remains to be determined. Notably, while the effect of each specific

gene in a hub may be tumor-specific, as evident from the differential effect of selected gCIS on different TNBC cell lines, the hubs identified in the mouse screens and the P-enriched, S- and M-enriched pathways identified in human breast cancer are consistently elevated (4 independent cohorts), and thus amenable to therapeutic intervention. These observations have direct implications for therapy. Targeting P-drivers may not only be futile but may even inadvertently promote metastasis. In contrast, combination therapies that target two or more S-drivers or S- plus M-enriched pathways may prove most effective in preventing metastatic disease.

RhoA signaling was consistently elevated in metastatic vs primary breast cancer, and in HER2-enriched, luminal A and luminal B subtypes, and may therefore be prioritized for therapy in combination with inhibitors of subtype-specific S-drivers to prevent metastatic disease. It is also induced in basal-like breast cancer metastases in the test cohort, but could not be validated due to the small number of samples. PI3K signaling was significantly higher in metastatic breast cancer of all subtypes in the four cohorts, and reproducibly in luminal A and luminal B. Both Rho GTPase and PI3K pathways promote primary tumor proliferation and survival[111,112], but our results reveal that these pathways are further enriched in metastases and are thus ideal for therapeutic targeting together with S-drivers. We previously demonstrated robust cooperation between inhibitors of CDC25 and PI3K in restricting growth of TNBC cells in culture and xenotransplantation models[83], but impact on metastatic disease is yet to be established. Further genomic and proteomic analysis in combination with additional clinical cohorts of paired primary tumors/metastases will likely uncover additional targetable genes and pathways that promote metastatic disease in breast and other cancers.

We demonstrated that high RhoA signaling combined with RB-loss identified TNBC patients with unfavorable prognosis in three different clinical databases. Such cooperation was not seen between high RhoA signaling and high PI3K, β-catenin or MET signaling, but was observed in patients with both RB-loss and high MET signaling. Thus, TNBC patients with RBKO-MET-high (S- plus S-) or RBKO-RhoA-high (S- plus M-) pathway activation should be identified and prioritized for therapy.

Rho GTPases are major accomplices in the hybrid ameboid - epithelial-to-mesenchymal transition state, which is driven by both EMT and Rho signaling, and affects a subset of tumor cells with high dissemination and metastatic potential[113,114]. In addition to RHO signaling and cell migration, metastatic-specific alterations may promote other unique features and plasticity of the metastatic cascade including partial epithelial-mesenchymal transition (EMT)/mesenchymal-epithelial transition (MET), anoikis-resistance, oxidative-phosphorylation vs glycolytic metabolic states, dormancy vs slow vs fast cycling, ROS/antioxidant regulation, lysyl oxidase (LOX) level, angiogenesis, and escape from immune-surveillance[6]. Identification of oncogenic alterations or cues from the tumor microenvironment that drive such metastatic-specific signaling and plasticity may uncover new druggable targets for combination therapies together with S-drivers.

In preliminary analysis for mutations and CNAs in 42 genes associated with Rho/Rac1/CDC42 signaling using paired primary-metastatic samples, we identified significant enrichment in missense mutations in PDGFR and RHOA and copy number gain in PAK1. Although most tumors do not exhibit increased oncogenic alterations in this pathway relative to primary lesions, RhoA signaling is significantly enriched in all metastatic breast cancer subtypes, suggesting that it is either induced by other mechanisms and/or that its sustained signaling is required for metastasis. Pharmacological inhibition of downstream factors such as Arp2/3, a major regulator of actin assembly and migration, may block RHO signaling regardless of the upstream oncogenic alterations or signals from the tumor microenvironment that may induce the pathway, and restrict metastatic dissemination. Indeed, using the Arp2/3 inhibitor CK666, we demonstrated cooperation with inhibitors of two

S-drivers: RB-loss and MET signaling in suppressing cell survival and migration. Targeting two different subtype-specific S-drivers or S- plus M-enriched drivers such as RHOA and PI3K pathways may effectively prevent metastatic dissemination and distal recurrence. The exact benefit from each combination regimen would likely reflect the efficacy of each drug and its propensity for synergy with other inhibitors, and should be comprehensively explored in future studies. In summary, our analysis uncovered drivers and pathways that promote metastatic breast cancer and suggest that distinct S-drivers cooperate with compartment-specific networks to promote local versus distal growth, with direct implications for prognosis and treatment of metastatic breast cancer.

## Methods

### Ethics and inclusion statement
All mouse experiments were performed in compliance with the Canadian Animal Care Council guide for the care and use of laboratory animals, and were approved by the Toronto General Hospital Research Institute Animal Research Committee at UHN to EZ. Mice were maintained on standard chow (Harlan-TEKLAD LM-4857912.15) at 20–24 °C, 36–40% humidity and 12 h dark/light cycle, and were routinely tested to ensure they are pathogen-free. Mice were also regularly monitored for signs of morbidity; no weight loss was observed in mice with lung metastasis. Lenti- and Adeno-virus work was conducted in accordance with the UHN Research Biosafety Manual with Biosafety Certificate issued to EZ at the Toronto General Research Institute, UHN.

### Mouse models
MMTV-Cre:Rb[f/f]:T2/Onc3a:R26[lsl_SB11] and MMTV-Cre:Rb[f/f]:T2/Onc3b:R26[lsl_SB11] compound female mice were generated by genetic crossing and genotyped by PCR analysis of tail biopsies using the following primers:

**Cre:**
EZ376, 5′-TCG CGA TTA TCT TCT ATA TCT TCA G
EZ377, 5′-GCT CGA CCA GTT TAG TTA CCC
**Rbf/+:**
EZ404, GGC GTG TGC CAT CAA TG;
EZ405, CTC AAG AGC TCA GAC TCA TGG.
**RbΔf:**
EZ404, GGC GTG TGC CAT CAA TG;
EZ445 plus 5′-GAAAGGAAAGTCAGGGACATTGGG-3′
**T2Onc3a:**
EZ621, 5′-TCACAATTCCAGTGGGTCAG-3′;
EZ622, 5′-TTTCATCATCGGCTGAACTG-3′
**T2Onc3b:**
EZ626 (12775wt L), 5′-GTGATGGGAGATGGAAATGG-3′
EZ627 (12775wt R), 5′-TGCTTACCCATCTCCAACCT-3′
EZ628 (12775onc3 L), 5′-AACTTTATCCGCCTCCATCC-3′
**ROSA26-LSL-SB11 (Cre-dependent sleeping beauty transposase):**
EZ623 – 5′-CACTTGCTCTCCCAAAGTCGCT-3′;
EZ624(reverse) – 5′-GGGGTGGTGATATAAACTTGAGGCT
EZ625(reverse) – 5′-GGCGGATCACAAGCAATAATAACC

SB mice were monitored for primary mammary tumor formation over a year period. When primary tumors reached 2 cm in diameter, tumor biopsies and macroscopic lung metastases were subjected to histology and DNA extraction, as described[33,38], followed by insertion analysis (see below). Fbxw7[f/f] mice[80], obtained from JAX laboratory (strain #:017563), were genotyped as recommended. Pten[f/f] mice were genotyped as described[82,115].

### Transplantation assays
Immune-deficient NSG female mice, 6–8 weeks of age, were injected with $1 \times 10^6$ MDA-MB-436 cells stably transduced with lenti-shRNA for FBXW7, CDC42BPA, MTMR3, or control scrambled DNA, mixed vol:vol with Matrigel, into the inguinal fat pad of NSG female mice (6 mice per group), as described[36]. At end points, control or test groups were sacrificed, tumor resected and weighted. Endpoints were 1.5 cm in diameter for primary tumor-only experiments, and 2 cm in diameter for experiments involving lung metastasis. Lungs were fixed in 10% formalin, embedded, sectioned in 50µ increments, subjected to hematoxylin & eosin (H&E) staining and scored for lung lesions using Image J 1.53a. Immunostaining of paraffin embedded tumor sections was performed as described[33,116].

### SB insertion sequencing Shear-SPLINK
A Covaris S220/E220 Focused-ultrasonicator (Covaris Inc., USA) was used to shear 100 µL of each DNA sample with parameters: peak incident power (W) – 140, duty factor – 10%, cycles per burst – 200, treatment time – 80, temperature 7 °C, water level – 12 cm. Epicenter End repair kit (Lucigen Corporation, USA) was used with 20 µL of Sonicated DNA, 0.5 µL ddH$_2$0, 3 µL kit buffer, 3 µL dNTP, 3 µL ATP and 0.5 µL kit enzyme mix. Sample was incubated at RT for 45 min and then 10 min at 70 °C. Linker+ and linker- primers (100 µM) were mixed at 1:1 ratio in Sodium-Tris-EDTA buffer (50 mM NaCl, 10 mM Tris-Cl - pH 8.0, 1 mM EDTA - pH 8.0). Primer solution was heated to 95 °C for 5 min and slowly cooled to room temperature. Fast-link ligase kit (Lucigen Corporation, USA) was used with 30 µL end-repaired DNA, 1.75 µL ATP, 1.64 µl adapter mix, 0.5 µL kit buffer, and 1.11 µL Fast-Link ligase. Solution was incubated at RT for 45 min and then the enzyme was inactivated with an incubation at 70 °C for 15 min. 35 µL the adapter ligation solution from previous step, 1 µL High Fidelity (HF) BamHI, 1.5 µL NEB buffer 4, 5 µL 10X bovine serum albumin (BSA), and 4 µL ddH$_2$0 were incubated overnight at 37 °C.

Two primary PCR reactions were set up for each side of the SB transposons (IRR and IRL). 5 µL DNA mix from previous step, 12.25 µL ddH$_2$0, 5 µL 5x Phusion buffer, 0.75 µL 10 mM MgCl$_2$, 0.5 µL 10 mM dNTPs, 0.5 µL 10Mm IRR or IRL primer, 0.5 µL 10Mm Linker-A1 primer, and 0.5 µL Phusion Taq (Sigma, USA). The sample was run using the following PCR cycle protocol: (1) 98 °C (30 s), (2) 98 °C (20 s), (3) 55 °C (30 s), (4) 72 °C (60 s), Steps 2, 3, 4 repeated 25 times, (5) 72 °C (60 s), (6) 4 °C (hold). 3 µL of the primary PCR was diluted 1:50, vortexed and incubated at RT for 30 min. PCR mix was made with 4 µL DNA mix from previous step, 32.5 µL ddH$_2$0, 10 µL 5x Phusion buffer, 1 µL 10 mM dNTPs, 2 µL 2.5 µM IR-barcoded transposon primer, 0.25 µL 10 µM Linker-A2 primer, and 1 µL Phusion Taq. Touch down PCR cycling protocol was used: (1) 98 °C (180 s), (2) 95 °C (30 s), (3) 49 °C (30 s), (4) 72 °C (60 s), Steps 2,3,4 repeated 10 times, (5) 95 °C (30 s), (6) 53.3 °C (60 s), (7) 72 °C (120 s), Steps 5,6,7 repeated 25 times, (8) 72 °C (60 s), (9) 4 °C (hold). The PCR products ran on the same lane were pooled and purified using Qiagen purification kit and resuspended in 50 µL TE buffer. A Nanodrop was used to determine the concentration of purified DNA. A maximum of 96 samples were pooled together from the IRL and IRR libraries per lane with a final concentration of 20–25 ng/µL. This pool was incubated at 40 °C for 30 min and submitted for sequencing on the Hiseq (Illumina, USA) paired-end 2 x 126 bp.

### SB read pre-processing, alignment, and analysis
Adapters were trimmed with cutadapt (v1.8) with parameters '-m 5–no-indels–discard-untrimmed -g R1_5prime = ^NNNNNNNNNTGTATGTAAACT TCCGACTTCAACTG' from read 1 (R1) for each sample. Since the SB insertions recognize and insert into a TA dinucleotide, only reads starting with a TA were kept for downstream steps. R1 reads were then paired with their respective paired reads (R2) and aligned with novoalign (v3.05.01) using parameters '-r ALL 1 -R O -c 8 -o SAM' with the mm9 mouse genome assembly. Aligned sam files were converted to bams for downstream analysis. Each integration address was annotated using the refFlat tables from UCSC genome database. Using the

chromosomal address the following information was extracted: [tumor ID], [gene name], [region of gene hit (e.g. intron, exon, and promoter)], [predicted affect of insertion on the expression of the gene], [number of reads on this insertion site within the sample], [orientation of the transposon relative to the gene]. Some insertion events were not annotated because they did not occur within a known gene. The IRL and IRR libraries are then merged together. If an insertion was detected in both libraries (i.e transposon orientations) the read and higher read count was used in the merged file. A dynamic filter was used to categorize the insertions as clonal or subclonal. For each library three thresholds were calculated using the insertion data: (i) >95% percentile of reads under the negative binomial distribution, (ii) 1% of the most abundant insertion sites, (iii) 0.1% of the total reads. The most stringent value was the threshold for clonal insertions, the second-most was the threshold of the clonal/subclonal category. Gene centric common insertion site (gCIS) analysis[42], was ran on each cohort using the clonal and subclonal/clonal insertions. This test was repeated for every gene and then $p$ values were adjusted using a stringent Bonferroni group-wise correction. Corrected $p$-values < 0.05 were called significant. The spindle protein gene Sfi1 was identified in both primary and metastatic lesions; this gene is frequently observed in SB screens due to its high copy number in the genome[19], and was omitted from the analysis.

## Bioinformatic analysis

**Datasets.** EGAS00001001753 - METABRIC 1904BC (Fig. 4a, b, Fig. 6g, Fig. S2A and Fig. S10F bottom)[85]; EGAS00001001753 - METABRIC 1794BC (Fig. 5c) was modified by removing BCs with a P53 pathway activity value between 0.4–0.5. METABRIC 205TNBC (Figs. 4c–f, 7c and Fig. S3), a TNBC dataset collected in our early work from a METABRIC breast cancer dataset[70]. GSE31519 – 383TNBC (Rody et al., 2011) (Fig. 4g). GSE81540 - SCAN-B 327TNBC (Fig. 4h, Fig. 7c, Fig. S3), and 3273BC (Fig. S10E)[117]. TCGA - TCGA1080BC (Supplementary Fig. S10F top)[118]. NODE OEP000155 - FUSCC360TNBC (Fig. 7c, Fig. S3)[119]. CCLE2019 (Fig. 4I). 58 breast cancer cell lines from 1,072 cell lines of multiple cancer types[73]. GSE81954 (97Pri-BC)[120] and GSE56493 (120Met-BC)[121] (Fig. 7a, b, Supplementary Fig. S9A top) are unpaired Pri and Met BC datasets from the Karolinska Institute. GSE147322 – 83Paired-MetBC (Fig. 7b Luminal A Validation cohort, Supplementary Fig. S9A bottom, and Fig S9B)[88]. GSE57968 – 36Paired-MetBC (Fig S9C top)[98]. GSE184869 – 45Paired-MetBC (Fig S9C bottom)[99]. MSK2018[103] and MSK2022[104] (Supplementary Fig. S10B–E) - The MSK2018 dataset with both primary and metastatic breast tumor specimens was collected between April 2014 and March 2017. The MSK2022 dataset with only metastatic breast cancers was collected between April 2014 and March 2020. The samples and patients collected before March 2017 were removed from the MSK2022 dataset, so that only unique specimens and patients from primary breast cancers in the MAS2018 and the metastatic breast cancers in both MSK2018 and MSK2022 datasets were analyzed. Kaplan–Meier curves were generated as described[70,71] or using kmplot.com.

**Pathways and PMut prediction.** PMut prediction was performed as described[122]. The MET pathway is as reported[68]; other pathways are as described in ref. 71. The 42 Rho/Rac/CDC42-pathway associated genes were assembled from the following links:

BIOCARTA_RHO_PATHWAY:

https://www.gsea-msigdb.org/gsea/msigdb/cards/BIOCARTA_RHO_PATHWAY

BIOCARTA_RAC1_PATHWAY:

https://www.gsea-msigdb.org/gsea/msigdb/cards/BIOCARTA_RAC1_PATHWAY

BIOCARTA_CDC42RAC_PATHWAY:

https://www.gsea-msigdb.org/gsea/msigdb/cards/BIOCARTA_CDC42RAC_PATHWAY

## Cell lines and cultures

Human breast cancer cell lines: MDA-MB-231, MDA-MB-436, MDA-MB-468, HCC38, Hs57T and MCF7 were maintained in DMEM containing 10% FBS and 1% PEST, at 37°C with 5 % $CO_2$. MDA-MB-436, and MDA-MB-231 were kindly obtained from the late Dr. Mona Gauthier, and the remaining were purchased as previously described[36,83], from the American Type Culture Collection (ATCC). Human embryonic kidney cells, HEK293T, obtained from Dr. Jason Moffat[123], were cultured as above.

## Stable shRNA knockdown cell lines

For gene knockdown, pLKO.1-puro-CMV plasmids containing shRNA for SB genes were purchased from Sigma-Aldrich Canada. Empty or non-target-puro shRNA control (cat. SHC016). or scrambled pLKO.1-puro-CMV plasmid was employed as a control. TRC numbers of each clone were: RB1 (TRCN0000288710) CDC42BPA (TRCN0000196639, TRCN0000194939, TRCN0000000659, TRCN0000199935, TRCN 0000196893), SRGAP2 (TRCN0000047958, TRCN0000047959, TRCN0000047961, TRCN0000047962), MTMR3 (TRCN0000003014), FBXW7 (TRCN0000235421, TRCN0000355644, TRCN0000368359, TRCN0000006558, TRCN0000235422), FBXW4 (TRCN0000010892, TRCN0000012819), and WDR33 (TRCN0000074839, TRCN00000 74840, TRCN0000074842, TRCN0000425945, TRCN0000435533). Lentiviral plasmids were expanded in *Escherichia coli*, and extracted using miniprep (Qiagen). For lentivirus production, packaging plasmids psPAX2 andPMD2, and with target vector, were co-transfected into HEK293T cells using PEI. Forty-eight hours post transfection media supernatant was harvested, passed through a 0.45 μm filter and then used to infect target cells. After 24 h infection, medium was changed, and cells either sorted for GFP positive cells and/or were grown in presence of puromycin to obtain resistant cells.

## Cell growth and in vitro wound assays

Cells were seeded in 96-well plates at $3–4 \times 10^3$ cells/well. In each day during the 4 days period, 30 μl of 2 mg/ml of MTT (3-[4,5-dimethyl-thiazol-2-yl]-2,5-diphenyltetrazolium bromide, Sigma) was added into each well and plates were incubated at 37 °C for 3 h. MTT/media was removed prior to adding 100 μl DMSO. The optical density (OD) was measured at 570 nm by a 96-well microplate reader (Molecular Devices). Assays were performed in 3–6 replicas and repeated at least 3 times. Hemocytometer cell counting performed each day during the 4 days period as well. For in vitro wound assay, cells were seeded in 6-well plates at 100% confluence to form a monolayer next day. A p1 pipet tip was used to create a scratch of the cell monolayer and wells were then washed and replaced a culture media containing 10 μg/ml mitomycin C or 500 nM aphidicolin. Migration progress was determined by taking snapshot under bright-field microscopy in indicated incubation times. Wound area was calculated using Image J.

## Incucyte proliferation and scratch-wound assays

**Proliferation.** Cells were seeded into flat bottom 96-well plates (Sarstedt 83.3924); the next days, drugs, made in media at double the intended concentration, were added. Treated plates were transferred into IncuCyte®ZOOM and maintained at 37 °C/5% $CO_2$. Phase contrast images of each well were taken every 4 h for 3 days at 10× objective to measure confluence over time. Confluency of each well was normalized to initial image to calculate fold changes over time.

**Scratch wound.** Cells were seeded into 96-well flat bottom Incucyte®Imagelock plates (Corning/Sartorius BA-0457) at 100% confluency the night prior to treatment. Drugs in media at 1X concentration were prepared the day earlier and frozen at −80 °C. The next day, scratch wounds were made in each well by IncuCyte®WoundMaker (Sartorius BA-04858) and each well washed 2 times with media. Media in wells were again aspirated and replenished with 200 μl of thawed media/1Xdrug. Plates were then imaged by IncuCyte®S3 housed in an

incubator maintained at 37 °C/5% $CO_2$. Using the Scratch-Wound Analyzer, phase contrast images of each well were taken every 2 h for 1–3 days at 10× objective wide mode to measure Relative Wound Density (RDW) over time.

## Western blot analysis

For western blot (WB), cells were washed with PBS and lysed using RIPA lysis buffer (0.15 M NaCl, 1% Nonidet P-40, 0.5% sodium deoxycholate, 0.1% SDS, 25 mM Tris 7.4, 5 mM NaF, 0.5 mM Na3VO4, and 1:100 protease inhibitor cocktail [1 mg/mL leupeptin, 2 μg/mL aprotinin, and 100 mM PMSF]). Protein concentration was determined by Pierce Reagent (Thermo Scientific). About 20 μg of protein was fractionated by SDS-PAGE and transferred onto PVDF membranes. The membranes were blocked with 5% nonfat dried milk in phosphate-buffered saline containing 0.05% Tween 20 (PBST) at RT for 1 h, washed 3 × 5 min with PBST, and incubated at 4 °C overnight with primary antibodies. Membranes were washed with PBST buffer 3 × 5 min each and incubated with HRP-conjugated anti-rabbit IgG secondary antibody (Cell Signaling) for 2 h. Primary antibodies: Rabbit anti-human RB1 (Cell Signalling Technologies, cat. 9313); rabbit anti-FBXW7 (EAP3553, Elabscience), rabbit anti-SRGAP2 (GTX130797, GeneTex), mouse anti-CDC42BPA (MRCKα) (sc-374568, Santa Cruz Biotechnology), mouse anti-MTMR3 (sc-393779, Santa Cruz Biotechnology), anti-pMLC2-Thr18/Ser19 (#3674), total MLC2 (#3672; Cell Signaling Technology), MRPL37 (ABcam 224467), GAPDH (sc-47724), rabbit anti-Tubulin (#2148, Cell Signaling Technology), and mouse anti-Actin (JLA20, Developmental Studies Hybridoma Bank). Secondary antibodies: anti-rabbit IgG-HRP (Cell Signalling Technologies, cat. 7074), anti-mouse IgG-HRP (Cell Signalling Technologies, cat. 7076). Primary antibodies were diluted 1:1000 in PBS (GAPDH 1:2000) with 5% BSA; secondary antibody was diluted 1:2000. Blots were imaged using ThermoFisher PicoPlus enhanced chemiluminescent (ECL).

## Statistics and reproducibility

Unless otherwise noted, cell culture experiments were performed in triplicates and three biological replicates. Data are presented as mean ± standard deviation (SD). Differences in gene expression and pathway activity were assessed by paired or unpaired $t$-test as indicated, and $P$ values calculated by two-tailed student $t$ tests using excel or PRISM 9 GraphPad Software (GraphPad Software, La Jolla, CA, USA) analysis. $P$ values for survival curves were determined by Log-rank (Mantel–Cox) test. For multiple comparison analysis, $P$ valued were determined using one-way ANOVA, Tukey's multiple comparison test. Differences in mutation and CNA frequency was analyzed by Fisher's exact test.

## Reporting summary

Further information on research design is available in the Nature Portfolio Reporting Summary linked to this article.

## Data availability

Sequence data from the SB insertional mutagenesis screens described in this study were deposited in the Gene Expression Omnibus (GEO) database under accession code GSE232167 Datasets analyzed herein are:     EGAS00001001753     [https://ega-archive.org/studies/ EGAS00001001753 /| https://www.cbioportal.org/study/summary?id= brca_metabric (version 2019)] s. GSE31519. GSE81540. TCGA [https:// www.cancer.gov/ccg/research/genome-sequencing/tcga].     NODE OEP000155.   CCLE2019   [https://depmap.org/portal/download/all/]. GSE81954. GSE56493. GSE147322. GSE57968. GSE184869. MSK2018 [https://www.cbioportal.org/study/summary?id=breast_msk_2018] [https://www.cbioportal.org/study/summary?id=breast_ink4_msk_ 2021]. Source data (raw data and uncropped images) are provided with this paper. Source data are provided with this paper.

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

## Acknowledgements

We thank Dr. Jiao Zhang for help in depositing the SB sequence data in GEO. This research was funded by grants from the Canadian Institute of Health Research (CIHR) and Canadian Breast Cancer Foundation (CBCF) to E.Z.; DoD USA army (BC150403) to S.E.E. and E.Z.; and Terry Fox Research Institute Program to J.R.W., S.E.E., M.D.T. and E.Z. C.M.P. was supported by funds from NCI RO1-CA148761 and Breast Cancer Research Foundation.

## Author contributions

Z.J. performed experiments, analyzed data and wrote the manuscript; Y.J., A.A., P.E.D.C., M.S., H.L., I.V., R.G.A. and E.M. performed experiments and analyzed results; P.S., D.W. and J.C.L. conducted bioinformatic analysis; A.D. provided S.B. mice and advice; Z.J., M.K., Y.B.D., J.R.W., C.M.P., G.D.B., S.E.E. and M.D.T. advised and supervised the analysis; E.Z., supervised and coordinated the analysis, analyzed data and wrote the manuscript with input from all authors.

## Competing interests

C.M.P. is an equity stockholder and consultant of BioClassifier LLC; C.M.P. is also listed as an inventor on patent applications for the Breast PAM50 Subtyping assay. There is no direct relationship between these PAM50 patents and the Intellectual Property and content of this study. The other authors declare no conflict of interest.

## Additional information

[1]Toronto General Research Institute - University Health Network, 101 College Street, Max Bell Research Centre, suite 5R406, Toronto, ON M5G 1L7, Canada. [2]Laboratory Medicine & Pathobiology, University of Toronto, Toronto, ON, Canada. [3]Program in Developmental & Stem Cell Biology Program, The Hospital for Sick Children, Toronto, ON, Canada. [4]The Arthur and Sonia Labatt Brain Tumour Research Centre, The Hospital for Sick Children, Toronto, ON, Canada. [5]The Donnelly Centre, University of Toronto, Toronto, ON, Canada. [6]Princess Margaret Cancer Center, University Health Network, Toronto, ON, Canada. [7]The Key laboratory of Chemistry for Natural Products of Guizhou Province and Chinese Academic of Sciences, Guiyang, Guizhou 550014, China. [8]State Key Laboratory for Functions and Applications of Medicinal Plants, Guizhou Medical University, Guiyang 550025, China. [9]Lunenfeld-Tanenbaum Research Institute, Sinai Health System, 600 University Avenue, Toronto, ON, Canada. [10]Lineberger Comprehensive Cancer Center, Departments of Genetics and Pathology, University of North Carolina, Chapel Hill, NC 27599, USA. [11]Department of Pathology, Carver College of Medicine, The University of Iowa, Iowa City, Iowa 52242, USA. [12]Department of Molecular Genetics, University of Toronto, Toronto, ON, Canada. [13]Department of Medicine, University of Toronto, Toronto, ON, Canada. [14]These authors contributed equally: YoungJun Ju, Amjad Ali, Philip E. D. Chung, Patryk Skowron, Dong-Yu Wang. ✉e-mail: eldad.zacksenhaus@utoronto.ca

