## [Peer Review File · Nature Communications]

REVIEWER COMMENTS

Reviewer #1 (Remarks to the Author): Expert in in vivo transposon mutagenesis screening and cancer functional genomics

In this study by Jiang et al, the authors perform a mammary specific Sleeping Beauty (SB) mutagenesis screen to investigate the relationship between oncogenic drivers that promote metastatic versus primary breast tumors. The screen was performed in treatment-naïve animals and in the context of Rb loss of function, which is frequently lost in breast cancers. The investigators identified common insertion site (CISs) in primary tumors (P-drivers), metastatic tumors (M-drivers), and in primary and metastatic tumors (shared-S drivers). The screen is very well-done, and the work is extensive, however most of the identified genes and pathways are not novel. Multiple Sleeping Beauty breast cancer screens have previously been published. As the investigators did not emphasize the novelty of their results and distinguish their work from previously published SB screens, the current study is not well suited for publication in Nature Communications. Major and minor points should be addressed and will improve the overall study:

1. It would be useful to know what fraction of breast cancers are diagnosed as metastases. This could be added to the introduction.
2. It is stated on p.3 (line 124-126) that no report has demonstrated clonal relationships between primary and metastatic gCISs. This might be true for breast cancer, however this type of analysis has been performed in other Sleeping Beauty screens for different tumor types (liver cancer), for example in Keng et al. Nature Biotechnology, 2009. The authors should clarify/modify this statement.
3. In the current study, the authors claim that RhoA and PI3K pathways are elevated in metastatic versus primary lesions, representing metastasis-enriched oncogenic pathways. In contrast, many studies suggest that RhoA is involved in all stages of tumor progression. The authors should acknowledge this discrepancy and speculate/provide commentary regarding why RhoA signaling is specific to breast cancer metastasis.

Minor points:

1. Figure 5 g,h,i are all mis-labeled as Figures 6 g,h,i (page 10, lines 423-425).
2. The following sentence is repeated twice on p. 12, "Induction of RhoA signalling in metastatic breast cancer could be caused by other genetic and epigenetic alternations, post-translational modification, or by external cues from the tumor microenvironment; their identification would be instrumental for precision therapy". Lines 541-544 and 545-547.

Reviewer #2 (Remarks to the Author): Expert in breast cancer functional genomics

The authors of this manuscript performed Sleeping-beauty mutagenesis in a murine model of breast cancer with loss of Rb to identify genes that drive primary or metastatic tumor growth or both. Using this approach they identified the MET-RAS pathway as shared driver, while Rho signaling, ubiquitination and RNE processing is metastasis-only driver. The authors followed up more detailed characterization of selected screen hits including Fbxw7 and Pten by generating double knock out mice or knock down in human breast cancer cell lines by shRNA.

The mouse model MMTV-Cre:Rbf/f mice has been previously described, Sleeping-beauty mutagenesis screens have also been performed for mammary tumors in mice. Thus, novelty is limited, and the genes/pathways identified have well known roles in breast tumorigenesis.

The manuscript is very dense (both the text and content) making it hard to read.

Specific comments:

1. Mammary tumors in mice are virtually always hormone receptor negative basal subtype, mostly reflecting one subset of triple-negative breast cancer (TNBC).
2. Rb loss is not that common in human breast tumors, it mainly occurs in a subset of TNBC or treatment-resistant ER+ luminal tumors (especially if treated with CDK4/6 inhibitors).
3. If the lung tumors are true metastases, then they should be derived from the primary mammary tumors and therefore should be clonally related – including shared transposon mutations. On the other hand, if the Rb deleted tumors are metastatic on their own w/o any additional mutation, then this is not really a screen to identify drivers of metastasis, but just drivers of tumor growth.
4. The follow up validation experiments using selected genes did not actually show enhanced metastasis with any of the genes tested, just increased tumor growth. Thus, they did not really identify any metastasis drivers.
5. The mechanistic studies are limited to cell growth and migration/invasion assays, which are very limited superficial characterization of cellular phenotypes w/o giving any mechanistic insights.

REVIEWER COMMENTS

Reviewer #1 (Remarks to the Author): Expert in in vivo transposon mutagenesis screening and cancer functional genomics

In this study by Jiang et al, the authors perform a mammary specific Sleeping Beauty (SB) mutagenesis screen to investigate the relationship between oncogenic drivers that promote metastatic versus primary breast tumors. The screen was performed in treatment-naïve animals and in the context of Rb loss of function, which is frequently lost in breast cancers. The investigators identified common insertion site (CISs) in primary tumors (P-drivers), metastatic tumors (M-drivers), and in primary and metastatic tumors (shared-S drivers). The screen is very well-done, and the work is extensive, however most of the identified genes and pathways are not novel. Multiple Sleeping Beauty breast cancer screens have previously been published. As the investigators did not emphasize the novelty of their results and distinguish their work from previously published SB screens, the current study is not well suited for publication in Nature Communications. Major and minor points should be addressed and will improve the overall study:

Response: We thank the Reviewer for suggesting that we emphasize the novelty of our results. Our results introduce several novel concepts and advances as summarized below:

- **To our knowledge, this is the first Sleeping Beauty (SB) mutagenesis screen in the mammary gland/breast cancer conducted on both primary tumors and metastases. All previous breast cancer screens were done only on primary lesions. This is a major leap forward as metastatic disease is the major cause of death from breast cancer.**
- **Another novelty of our work is the identification of oncogenic networks that drive metastatic breast cancer, and the demonstration of primary (P)-only, metastasis (M)-only**

and shared (S) oncogenic drivers that promote both primary and metastatic mammary tumors.

- Critically, we showed that a similar pattern of primary-enriched, metastasis-enriched and shared oncogenic drivers is observed in human breast cancer (4 different data sets). We found Rho signaling/motility as a major metastatic driver that is enriched both in our metastatic SB screen and in human metastatic breast cancer.
- TNBC patients with RB loss-MET-high (two S-drivers) or RB loss -RhoA high (S-driver plus M-driver) pathway activation exhibit exceedingly poor prognosis and should be prioritized for therapy.
- Inhibitors of S-drivers (RB-loss, MET) plus M-driver (Rho pathway) cooperate to suppress cell proliferation and migration.
- Our analysis supports the novel idea that S-drivers cooperate with compartment specific P- or M-drivers to promote local versus distal growth, respectively,
- Our results also suggest that “*targeting S- and M- but not P-drivers offers a rationale avenue to prevent metastatic breast-cancer.*” – we ended the revised Abstract with this sentence.
- Our SB screens reveal metastasis-specific drivers even in drug-naïve animals. Thus, new mutations found in metastasis-only breast cancer samples from human patients may not only be due to drug selection but also represent genuine metastasis-promoting drivers.
- Finally, we show that the M-drivers form specific interactomes (Fig. 2a) and pathways (Fig. 3a), and that components of these interactomes/pathways correlate with poor clinical outcome (Fig. 3c; supplemental Figs. S2, S3, S5). Thus, the Shared and Metastasis-specific drivers we have identified (Fig. 1, supplemental Excel Tables) provide a rich resource for future basic and translational analysis.

We emphasized all these points in the revised manuscript, e.g. Abstract, last sentence; Discussion, first paragraph.

1. It would be useful to know what fraction of breast cancers are diagnosed as metastases. This could be added to the introduction.

Response: Thanks - we have provided this important information in the revised Introduction (second line).

2. It is stated on p.3 (line 124-126) that no report has demonstrated clonal relationships between primary and metastatic gCISs. This might be true for breast cancer, however this type of analysis has been performed in other Sleeping Beauty screens for different tumor types (liver cancer), for example in Keng et al. Nature Biotechnology, 2009. The authors should clarify/modify this statement.

Response: We thank the Reviewer for pointing out this oversight on our part. We referenced the Keng et al. paper in the initial submission but did not credit it for demonstrating clonal relationship between primary and metastatic HCC. We have now revised the text as following “... only one report, on hepatocellular carcinoma, demonstrated clonal relationships between primary and metastatic insertion-sites (Ref 30 - Keng et al. Nature Biotechnology, 2009). The analysis we performed with multiple independent pairs of mice establishes such clonal relationship in breast cancer for the first time.

3. In the current study, the authors claim that RhoA and PI3K pathways are elevated in metastatic versus primary lesions, representing metastasis-enriched oncogenic pathways. In contrast, many studies suggest that RhoA is involved in all stages of tumor progression. The authors should acknowledge this discrepancy and speculate/provide commentary regarding why RhoA signaling is specific to breast cancer metastasis.

Response: We agree that Rho and PI3K signaling are elevated in primary tumors vs normal cells but our results show that these pathways are further and significantly induced in metastasis. The new statistical analysis on combined 4 cohorts (new Fig. 7b) clearly demonstrate that. We added

the following statement in Discussion (p15) to clarify this point “Both Rho GTPase and PI3K signalings promote primary tumor proliferation and survival (Ref: 108-109), but our results reveal that these pathways are elevated in metastases and are thus ideal for therapeutic targeting together with S-drivers”.

Minor points:

1. Figure 5 g,h,i are all mis-labeled as Figures 6 g,h,i (page 10, lines 423-425).

Response: Thanks- we fixed that – to new Fig. 5h-j.

2. The following sentence is repeated twice on p. 12, “Induction of RhoA signalling in metastatic breast cancer could be caused by other genetic and epigenetic alternations, post-translational modification, or by external cues from the tumor microenvironment; their identification would be instrumental for precision therapy”. Lines 541-544 and 545-547.

Response: Thanks - we removed this duplication.

Reviewer #2 (Remarks to the Author): Expert in breast cancer functional genomics

The authors of this manuscript performed Sleeping-beauty mutagenesis in a murine model of breast cancer with loss of Rb to identify genes that drive primary or metastatic tumor growth or both. Using this approach they identified the MET-RAS pathway as shared driver, while Rho signaling, ubiquitination and RNA processing are metastasis-only drivers. The authors followed up more detailed characterization of selected screen hits including Fbxw7 and Pten by generating double knock out mice or knock down in human breast cancer cell lines by shRNA.

The mouse model MMTV-Cre:Rbf/f mice has been previously described, Sleeping-beauty mutagenesis screens have also been performed for mammary tumors in mice. Thus, novelty is limited, and the genes/pathways identified have well known roles in breast tumorigenesis.

The manuscript is very dense (both the text and content) making it hard to read.

Response: We thank the Reviewer for suggesting that we emphasize the novelty of our results. Our results introduce several novel concepts and advances as summarized below:

- **To our knowledge, this is the first Sleeping Beauty (SB) mutagenesis screen in the mammary gland/breast cancer conducted on both primary tumors and metastases. All previous breast cancer screens were done only on primary lesions. This is a major leap forward as metastatic disease is the major cause of death from breast cancer.**
- **Another novelty of our work is the identification of oncogenic networks that drive metastatic breast cancer, and the demonstration of primary (P)-only, metastasis (M)-only and shared (S) oncogenic drivers that promote both primary and metastatic mammary tumors.**
- **Critically, we showed that a similar pattern of primary-enriched, metastasis-enriched and shared oncogenic drivers is observed in human breast cancer (4 different data sets). We found Rho signaling/motility as a major metastatic driver that is enriched both in our metastatic SB screen and in human metastatic breast cancer.**
- **TNBC patients with RB loss-MET-high (two S-drivers) or RB loss -RhoA high (S-driver plus M-driver) pathway activation exhibit exceedingly poor prognosis and should be prioritized for therapy.**
- **Inhibitors of S-drivers (RB-loss, MET) plus M-driver (Rho pathway) cooperate to suppress cell proliferation and migration.**

- Our analysis supports the novel idea that S-drivers cooperate with compartment specific P- or M-drivers to promote local versus distal growth, respectively,
- Our results also suggest that “*targeting S- and M- but not P-drivers offers a rationale avenue to prevent metastatic breast-cancer.*” – we ended the revised Abstract with this sentence.
- Our SB screens reveal metastasis-specific drivers even in drug-naïve animals. Thus, new mutations found in metastasis-only breast cancer samples from human patients may not only be due to drug selection but also represent genuine metastasis-promoting drivers.
- Finally, we show that the M-drivers form specific interactomes (Fig. 2a) and pathways (Fig. 3a), and that components of these interactomes/pathways correlate with poor clinical outcome (Fig. 3c; supplemental Figs. S2, S3, S5). Thus, the Shared and Metastasis-specific drivers we have identified (Fig. 1, supplemental Excel Tables) provide a rich resource for future basic and translational analysis.

We emphasized all these points in the revised manuscript, e.g. Abstract, last sentence; Discussion, first paragraph.

Finally, we made multiple changes in the manuscript to simplify the text, as requested.

Specific comments:

1. Mammary tumors in mice are virtually always hormone receptor negative basal subtype, mostly reflecting one subset of triple-negative breast cancer (TNBC).

Response: We discussed this issue with Chuck Perou, a co-author on this manuscript, who is an expert in breast cancer and molecular subtyping of human breast cancer and mouse models.

Virtually all mouse models are ER-negative, but not all are of the basal subtype. The following two references: PMID: 17493263 and PMID: 24220145 from Perou’s lab, show that most mouse models are not basal-like subtype as determined by molecular profiling. Although mouse models are ER-negative/ER-low, ER level is just one of many possible ‘luminal tumor’ markers, and the widely used MMTV-NEU and MMTV-PYMT are classified as luminal but are ER-negative/low. There do exist human tumors that are luminal and ER-negative/PR-negative, most would call these TNBC “LAR” subtype as these TNBC luminal tumors are often AR+.

Finally, the Perou and Zacksenhaus labs showed that Rb-negative lesions from our MMTV-Cre:Rbf/f mouse model are diverse, and cluster together with luminal, basal-like or mesenchymal-like/claudin-low like lesions (PMID: 20679727; PMID: 27571409).

2. Rb loss is not that common in human breast tumors, it mainly occurs in a subset of TNBC or treatment-resistant ER+ luminal tumors (especially if treated with CDK4/6 inhibitors).

Response: We thank the Reviewer for pointing out this issue, which we have now further clarified in the manuscript by adding a sentence and the references below in the Introduction.

As indicated in the manuscript, RB can be disrupted by specific mutations/deletions/epigenetic silencing of the gene or by phosphorylation of pRB via cyclin-dependent kinases. Phosphorylation dissociates pRB from its major targets such as E2F1-3, allowing cell cycle progression. The Perou’s lab/Cancer Genome Atlas Network showed in a 2012 Nature paper (PMID: 23000897) that the RB gene is mutated in 2% of luminal and 4% of basal-like BC (Fig. 1, *ibid*). However, the RB pathway is lost in ~20% of basal-like breast cancer through multiple additional mechanisms (e.g. RB mutation/silencing; cyclin E amplification; p16ink4a lost - Table 1, *ibid*). Using RB-loss/p53-loss signatures and oncoprint analysis of RB1 and TP53 alterations, we calculated that the two genes are lost in 28-40% of TNBC (JCI, 2016 - PMID: 27571409). Thus, while RB loss by nonsynonymous mutation is not that common, it is frequently inactivated by other mechanisms. Similar conclusion was presented in Nik-Zainal, S. et al (Nature, 2016; PMID: 27135926, Fig 1) in which RB1 loss is among the 9 most frequently altered genes in BC and second most altered in ER-negative BC. Another strategy for identification of cancer driver genes based on nucleotide context rather than nonsynonymous mutation (Nat Genet. 2020; PMID: 32015527)

pointed to RB (Fig 4) and RB pathway (Fig. 6) as one of the most frequently altered tumor suppressors in breast and other cancers. Finally, RB, TP53 and PI3K pathways are most often lost in metastatic cancer of all types (Nature 2017; PMID: 28783718).

In this manuscript, conditional Rb deletion in the mammary gland mimics the different mechanisms that inactivate the Rb gene (mutations/deletions/silencing) or pRb protein (hyper-phosphorylation) – and identifies oncogenic alterations that cooperate with such dysregulation of the cell cycle/Rb pathway to promote primary and metastatic disease. We conveyed this point on page 3 in the Introduction, and referenced the above papers on the frequency of RB1 loss in breast cancer.

3. If the lung tumors are true metastases, then they should be derived from the primary mammary tumors and therefore should be clonally related – including shared transposon mutations. On the other hand, if the Rb deleted tumors are metastatic on their own w/o any additional mutation, then this is not really a screen to identify drivers of metastasis, but just drivers of tumor growth.

Response: We suggest that because we were able to demonstrate clonality (Fig. 2b-d), the lung lesions represent true metastases and not independent primary lung tumors. Without demonstrating such clonal relationship, one cannot assume that “the lung tumors are true metastases”.

Regarding the second point, although the Rb pathway as well as TP53 and the PI3K pathways are often altered in metastases of diverse types of malignancies including breast cancer (PMID: 28783718), each alteration alone is not sufficient to promote primary and metastatic disease. They require cooperating oncogenic events, and our screen has identified such cooperating oncogenic drivers in the context of RB loss that promote both primary growth and metastasis.

4. The follow up validation experiments using selected genes did not actually show enhanced metastasis with any of the genes tested, just increased tumor growth. Thus, they did not really identify any metastasis drivers.

Response: We thank the reviewer for raising this issue. In the revised manuscript, we present new data to address this critique. Specifically, we determined the consequences of CDC42BPA and MTMR3 knock-down on both primary tumor growth and lung metastasis following orthotopic transplantation. We observed significant increase in lung metastasis in both cases (new Fig. 5i-j).

We note that we also show that M-drivers promote cell proliferation and/or migration when over-expressed or depleted in TNBC cells. Increased cell migration is metastasis-promoting activity and a hallmark of metastatic dissemination.

5. The mechanistic studies are limited to cell growth and migration/invasion assays, which are very limited superficial characterization of cellular phenotypes w/o giving any mechanistic insights. **Response:** We thank the reviewer for raising this issue as well. In the revised manuscript, we present new data to address the mechanism by which CDC42BPA, a M-driver, affects cell migration. Specifically, we determined the effect of CDC42BPA-depletion alone or together with the ROCK inhibitor Fasudil on phosphorylation of Myosin Light Chain 2 (MLC2) at Thr18/Ser19, which controls contractility, and on cell migration. We found that CDC42BPA modulates MLC2 phosphorylation and cell migration by antagonizing the effect of ROCK in TNBC cells (Fig. 5f and supplementary Fig. S7d-e). We note that for CDC42BPA and other representative M-drivers, we demonstrated they promote cell proliferation, migration and/or tumorigenesis and metastasis (new Fig. 5i-j); for FBXW4 we also identified several factors (BCI2 and MRPL37) as potential targets.

Reviewers' comments:

Reviewer #1 (Remarks to the Author):

The authors have adequately addressed my comments, although it would be useful to reference the following reviews regarding the state of in vivo SB mutagenesis screens:

Moriarity B and Largaespada D, Current Opinion in Genetics & Development,2015

O'Donnell K, Current Opinion in Genetics & Development,2018

Reviewer #2 (Remarks to the Author):

The authors have responded to each of the specific points raised by the reviewers and added some additional data, but the overall changes are rather limited and the issue of the genes identified are true “metastasis-only” drivers is not proven – but tumor driver vs. metastasis-drive distinction may not be very clear anyway.

Point-by-point response to the reviewers' comments:

Reviewer #1 (Remarks to the Author):

The authors have adequately addressed my comments, although it would be useful to reference the following reviews regarding the state of in vivo SB mutagenesis screens:

Moriarity B and Largaespada D, Current Opinion in Genetics & Development, 2015

O'Donnell K, Current Opinion in Genetics & Development, 2018

Response: We thank the reviewer for accepting our revisions – and for suggesting this reference, which we have incorporated into the revised manuscript (reference 18).

Reviewer #2 (Remarks to the Author):

The authors have responded to each of the specific points raised by the reviewers and added some additional data, but the overall changes are rather limited and the issue of the genes identified are true “metastasis-only” drivers is not proven – but tumor driver vs. metastasis-drive distinction may not be very clear anyway.

Response: The Reviewer raised two issues:

1. “the authors... added some additional data, but the overall changes are rather limited”
As the reviewer noted and as outlined below, we made all the revisions suggested by the Reviewer in the initial review:

a. “Sleeping-beauty mutagenesis screens have also been performed for mammary tumors in mice. Thus, novelty is limited,”

As we explained, the manuscript presents novel concepts and advancements with important implications to metastatic breast cancer progression and therapy:

- 1. The manuscript describes the first transposon-mediated mutagenesis and analysis of metastatic breast cancer. Given that metastases, not primary tumors, are the major cause of death from breast cancer, this manuscript is an important step forward.**
- 2. We demonstrated clonal relationship between primary and metastatic gCISs – and identified primary-specific, shared- and metastasis-specific pathways/hubs.**
- 3. We identified a large set of “metastasis-enriched genes/pathways” in drug-naïve animals that can be used as a unique and rich resource for future analysis by us and others. Many of these genes are already known or were shown by us in this manuscript to promote hallmarks of metastasis.**
- 4. We show that a similar organization of primary-enriched, shared and metastasis-enriched pathways are observed in primary human breast cancer and metastases from 4 independent cohorts. We further show that inhibition of shared plus metastasis-specific pathways synergizes to restrict cell proliferation and migration.**

- 5. Our results suggest that targeting shared and metastasis-enriched pathways, not primary-enriched pathways may offer a new modality to prevent metastatic disease. This is a novel and important finding given the tremendous efforts spent on developing drugs that target primary-enriched drivers, and the rationale provided here for targeting shared and metastasis-enriched oncogenic pathways.**

b. “The follow up validation experiments using selected genes did not actually show enhanced metastasis with any of the genes tested, just increased tumor growth. Thus, they did not really identify any metastasis drivers”.

- **We provided new data showing that knock-down of *CDC42BPA* and *MTMR3* increases lung metastasis following orthotopic transplantation (new Fig. 5i-j).**
- **Notably, we already showed that M-drivers promote cell proliferation and/or migration when over-expressed or depleted in TNBC cells. Increased cell migration is a metastasis-promoting activity and a hallmark of metastatic dissemination.**
- **We presented Kaplan-Meier curves that many of the affected genes are associated with poor clinical outcome (main and supplemental Figures).**
- **Finally, in the revised manuscript we referenced multiple papers showing that many of the gCIS identified in the metastasis-only hubs have been implicated in cancer progression/metastasis in other types of cancer (references 51-61).**

c. “The mechanistic studies are limited to cell growth and migration/invasion assays, which are very limited superficial characterization of cellular phenotypes w/o giving any mechanistic insights”.

- **First, our genetic analysis uncovered novel mechanisms of metastasis in mouse breast cancer, highlighting genes and pathways that drives metastasis in drug-naïve animals. Remarkably, we show that the same arrangement seen in the mouse - of P-, M- and shared gCIS - is also observed in human breast cancer and metastasis (Fig. 7). In the mouse screen, we identified genes that have previously been implicated in cancer progression/ metastasis (references 51-61). We further demonstrated that representative gCIS from the metastatic hubs promote cell migration, a hallmark of metastasis, in TNBC cells.**
- **Second - in response to this reviewer concern (point b above), we provided new data showing that some of these genes promote metastasis to the lung in xenotransplantation models.**
- **Third, in response to the concern about mechanistic insights – we presented new data to address the mechanism by which *CDC42BPA*, an M-gCIS, affects cell migration. We showed that *CDC42BPA* modulates *MLC2* phosphorylation and cell migration by antagonizing the effect of *ROCK* in TNBC cells (Fig. 5f and supplementary Fig. S7d-e).**

d. We fully addressed/clarified specific comments 1-3 by the Reviewer about the text/concepts in breast cancer.

These are extensive revisions that addressed each and every concern and suggestion by the Reviewer. The manuscript already contains an enormous amount of work including genetic screens of hundreds of mice, primary tumors and metastases, genetic crosses, biochemical, cell culture, transplantation experiments and bioinformatic analysis. We are ready to address other concerns, such as point 2 below.

2. “the issue of the genes identified are true “metastasis-only” drivers is not proven – but tumor driver vs. metastasis-drive distinction may not be very clear anyway.”

We thank the reviewer for raising this new but important issue. We used the term “metastasis-specific” gCIS/hubs based on the tissue in which these gCIS were identified (mammary gland or lung) using high stringent analysis at the filtered_clonal level. Other papers on SB screen on metastasis such as the liver specific SB screen (Nat Biotech 2009) used the terms “metastasis-specific insertion mutations” and “metastasis-associated genes”, which are similar to the nomenclature we use: “metastasis-specific gCIS”.

As noted under subsection - **A major network of shared-drivers, and metastatic-specific hubs** – on page 6 – “...insertions in *Plag1*, found at the filtered_clonal level to be primary-only (Fig. 1f; supplemental Table S1), were detected at the subclonal level in lung metastases, whereas insertions in *Fbxw4*, found only in lung metastases at the clonal level, were also detected in primary lesions at the subclonal level (supplemental Tables S3-4).”

Given the reviewer concern, we have now added the following sentence in the revised manuscript (p6): “Therefore, while our designation of P- and M-specific gCIS is based on filtered_clonal analysis, at the subclonal level, they can be denoted P- and M-enriched gCIS.”. This designation is also consistent with the results we obtained with human breast cancer where we observe by pathway analysis P-enriched and M-enriched (as well as Shared) oncogenic signaling.

We also made this point in the Discussion - p14.

Furthermore, we changed the title of the manuscript from

“Distinct shared and compartment-specific oncogenic networks drive primary versus metastatic breast cancer”

to

“Distinct shared and compartment-enriched oncogenic networks drive primary versus metastatic breast cancer”

We anticipate both metastasis-specific and primary-specific gCIS to be different in SB screen on an Rb-deficient background versus screens on other genetic backgrounds. Indeed, as expected, no gCISs were identified in our screens on the Rb pathway (e.g. Cdk or Cdk-inhibitors such as p16ink4a) as seen in human breast cancer (supplemental Fig. S1b-e) because Rb is already deleted in our screen. In addition, we do not expect *a priori* that tumorigenesis in the mouse to completely mirror the human disease since different gene family members may be dominant, and level of compensation (redundancy) may be different in the two species. For example, in the case of RB, germ line mutations in this tumor suppressor predispose to retinoblastoma in human but not in the mouse in which combined inactivation of other members of the Rb family (p107 or p130) or Cdk-inhibitors (e.g. p21) is required for tumorigenesis.

More importantly, we suggest that the effect of each specific gCIS is context- and tumor-specific, as evident from the differential effect of selected gCIS on different TNBC cell lines. On the other hand, the hubs identified in the mouse screens and the P-enriched, S and M-enriched pathways identified in human breast cancer are consistently elevated (4 independent cohorts), and thus amenable to therapeutic intervention. We made this point on page 15 in the revised manuscript. Clearly, oncogenic alterations of different genes on a pathway can activate or disrupt it as shown – for example – in Ref 35 from co-author Chuck Perou (Koboldt, D.C. et al. Comprehensive molecular portraits of human breast tumours. Nature 490, 61-70 (2012)). However, targeting a critical downstream component of a pathway can often inhibit the entire pathway irrespective of the specific upstream driving mutation, as we showed for example with the Arp2/3 inhibitor, CK666, downstream of RHO signaling.

The metastasis-enriched pathways include PI3K and RHO signaling, which are known to promote metastasis (Ref 9, 115-116). This is also consistent with our findings that components of RHO signaling such as CDC42BPA, SRGAP2 and WASF2 form one of the metastasis-associated hubs in the SB screen. Therefore, our analysis of human breast cancer parallels the results from the mouse screens with a similar organization and pathways (e.g. shared-pathways include MET oncogenic signaling; whereas metastasis-enriched pathways include RHO signaling - both in the mouse SB screens in mammary gland and lung, and in human breast cancer and metastases).

In summary, the mouse genetic screens uncovered (i) multiple genes that promote hallmarks of metastasis at least in some contexts/tumor cells and (ii) the principle of Primary-specific, Metastasis-specific and Shared gCIS, which prompted the analysis of human breast cancer and the discovery of a similar organization with Primary-enriched, Shared and Metastasis-enriched pathways (Fig. 7). Together, our analysis of both mouse and human metastatic breast cancer suggests that targeting Shared and Metastasis- but not Primary-enriched oncogenic pathways may offer a rationale avenue to prevent metastatic dissemination.

IN CONCLUSION, the designation of metastasis-specific (or -enriched, depending on the stringency of analysis) gCIS is based on the compartment (mammary gland or lung) in which they were detected. We provide compelling evidence that the gCIS from the metastasis-specific hubs promote cancer invasion/metastasis based on our analysis in TNBC and on published work. In line with the Reviewer note (– but tumor driver vs. metastasis-drive distinction may not be very clear anyway”) - this does not mean that some of these genes cannot act as primary-specific or shared oncogenic drivers in different contexts, in as much as “primary specific gCIS” may promote metastasis in some other contexts. But this is what we see on Rb-knockout background. Importantly, we then looked at human breast cancer and identified primary-enriched, shared and metastasis-enriched pathways in 4 different clinical cohorts. To the Reviewer point (“the genes identified are true “metastasis-only” drivers is not proven -) – the metastasis-enriched (and primary-enriched) pathways in human breast cancer can be seen as well-established / “proven” – and can thus serve as a the basis for therapeutic intervention.

--

REVIEWERS' COMMENTS

Reviewer #2 (Remarks to the Author):

The authors properly addressed all remaining criticism and the additional changes they made significantly improved the paper. The data should be all made publicly available to facilitate research in this area.